# Representing high-latitude deep carbon in the pre-industrial state of the ORCHIDEE-MICT land surface model (r8704)

Yi Xi[1], Philippe Ciais[1], Dan Zhu[2], Chunjing Qiu[3,4], Yuan Zhang[5], Shushi Peng[2], Gustaf Hugelius[6,7], Simon P. K. Bowring[1], Daniel S. Goll[1], Ying-Ping Wang[8]

[1]Laboratoire des Sciences du Climat et de l'Environnement, LSCE/IPSL, CEA–CNRS–UVSQ, Université Paris-Saclay, 91191 Gif-sur-Yvette, France
[2]Sino-French Institute for Earth System Science, College of Urban and Environmental Sciences, Peking University, Beijing, China
[3]Research Center for Global Change and Complex Ecosystems, School of Ecological and Environmental Sciences, East China Normal University, Shanghai, China
[4]Institute of Eco-Chongming, East China Normal University, Shanghai, China
[5]Key Laboratory of Alpine Ecology, Institute of Tibetan Plateau Research, Chinese Academy of Sciences, Beijing, China
[6]Department of Physical Geography, Stockholm University, 10691 Stockholm, Sweden
[7]Bolin Centre for Climate Research, Stockholm University, 10691 Stockholm, Sweden
[8]CSIRO Environment, Private Bag 10, Clayton South, VIC 3169, Australia.

*Correspondence to*: Yi Xi (yi.xi@lsce.ipsl.fr)

**Abstract.** Field measurements, after extrapolation, suggest that deep Yedoma deposits (ice-rich, organic-rich permafrost, formed during the late Pleistocene) and peatlands (formed during the Holocene) account for about 600 Pg C of soil carbon storage. Incorporating this old, deep, cold carbon into land surface models (LSMs) is crucial for accurately quantifying soil carbon responses to future warming. However, it remains underrepresented or absent in current LSMs, which typically include a passive soil carbon pool (a conceptual soil carbon pool with longest turnover time) to represent all 'old carbon' and lack the vertical accumulation processes that deposited deep carbon in the soil layers of peat and Yedoma. In this study, we propose a new, more realistic protocol for simulating deep and cold carbon accumulation in the northern high latitudes (30°N-90°N), using the ORCHIDEE-MICT model. This is achieved by 1) integrating deep carbon from Yedoma deposits whose formation is calculated using Last Glacial Maximum climate conditions, and 2) prescribing the inception time and location of northern peatlands during the Holocene using spatially explicit data on peat age. Our results show an additional 157 Pg C in present-day Yedoma deposits, as well as a shallower peat carbon depth (by 1-5 m) and a smaller passive soil carbon pool (by 35 Pg C, 43%) in northern peatlands, compared to the old protocol that ignored Yedoma deposits and applied a uniform, long-duration (13,500 years) peat carbon accumulation across all peatlands. As a result, the total organic carbon stock across the Northern Hemisphere (30°N-90°N) simulated by the new protocol is 2,028 Pg C, which is 226 Pg C higher than the previous estimate. Despite the significant challenge in simulating deep carbon with ORCHIDEE-MICT, the improvements in the representation of carbon accumulation from this study provide a model version to predict deep carbon evolution during the last glacial-deglacial transition and its response to future warming. The methodology implemented for deep carbon initialization in permafrost and cold regions in ORCHIDEE-MICT is also applicable to other LSMs.

# 1 Introduction

Soil organic carbon (SOC) is a major component of the global carbon cycle, as the size and composition of this reservoir directly control the amount of carbon dioxide ($CO_2$) emitted from terrestrial ecosystems via heterotrophic respiration (Rh) (Jackson et al., 2017). In the absence of other disturbances, the imbalance between the responses of Rh and net primary production (NPP) to climate warming determines whether terrestrial ecosystems will become a carbon source or a carbon sink in the future (Crowther et al., 2016; Lu et al., 2013; Walker et al., 2018). Earth system models (ESMs) employed in the recent IPCC Sixth Assessment Report (AR6) predict that terrestrial ecosystems will continue to act as a net carbon sink until 2100 due to the relatively dominant response of NPP to warming, but may shift to a net source after 2100 as SOC respiration exceeds vegetative carbon uptake (IPCC, 2021). Variations in SOC density also indirectly influences global carbon cycling processes by altering soil hydrology (e.g., soil porosity) and soil heat transfer (e.g., thermal diffusivity, especially in cold regions) (Zhu et al., 2019). Collectively, these suggest that accurately representing and estimating global SOC stocks is paramount to better understand the overall response of terrestrial ecosystems to future climate warming.

Widely-used global SOC stock gridded datasets such as HWSD (the Harmonized World Soil Database; Hiederer and Köchy, 2012), WISE30sec (the World Inventory of Soil property Estimates project; Batjes, 2016), and SoilGrids250m (global gridded soil information; Hengl et al., 2017), are currently based on site-level soil profile measurements, which are extrapolated to regional scales through their combination with digitized soil maps (HWSD and WISE30sec) or via machine learning techniques (SoilGrids250m). Due to the challenges posed by deep soil coring and sampling, these products usually only produce global SOC estimates to relatively shallow depths, e.g., 1,417 Pg C for 0-1 m soil depth from HWSD (Hiederer and Köchy, 2012) and 2,060 ± 215 Pg C for 0-2 m soil depth from WISE30sec (Batjes, 2016). However, high latitude soils underlain by permafrost and peat typically exhibit vertical profiles significantly deeper than 2 m. Accounting for SOC stocks at the 2-3 m depth range would add roughly ~470 Pg C (+23%) to the global soil stocks, including 175 (142-220) Pg C in permafrost deep soils (Mishra et al., 2021), 199 Pg C in non-permafrost deep mineral soils (Jobbagy and Jackson, 2000), and 92 Pg C in northern peatlands outside permafrost regions (Jackson et al., 2017).

Going deeper than 3 m, a substantial storage of SOC can still be found. In particular, in Alaska, northern Siberia and northwest Canada, there is an area of 0.48 million km$^2$ (hereafter Mkm$^2$) covered by icy, organic-rich, silt-dominated sediments, known as Yedoma (Strauss et al., 2017). These sediments were deposited to an average depth of 19-25 m by windblown particles over ice-free regions during the Pleistocene (Anthony et al., 2014; Strauss et al., 2013). Yedoma likely covered a vast area during the late Pleistocene, though only ~30% has survived thawing and warming during the transition to the Holocene (Strauss et al., 2017). Empirical extrapolations based on field measurements suggest a SOC storage of 115 (83-269) Pg C (Strauss et al., 2017) in present-day Yedoma deposits. Peatlands are another reservoir of deep organic carbon in the northern high latitudes. Most formed during the Holocene, following the drainage of thermokarst lakes fed by the thawed ice from Yedoma soils

(Anthony et al., 2014). Peat carbon can accumulate to depths exceeding 10 m, due to anaerobic constraints on decomposition (Hugelius et al., 2020). At present, northern peatlands cover an area of 3.7 (±0.5) - 4.0 Mkm$^2$ and account for 415 (±150) - 547 (±74) Pg C (Hugelius et al., 2020; Yu et al., 2010). Despite large uncertainties in these estimates, which result from inadequate field measurements and rough estimation methods, the substantial SOC storage (~600 Pg C) in Yedoma deposits and northern peatland soils highlights the importance of incorporating deep carbon in the accounting of global soil carbon stocks.

Land surface models (LSMs) are valuable tools for assessing the response of terrestrial ecosystems to climate warming. However, the simulated response by most LSMs is subject to biases in simulated SOC due to limitations in existing model structures and missing processes, e.g., deep carbon from Yedoma deposits and peatland soils (Bradford et al., 2016; Koven et al., 2015; Niu et al., 2024; Shi et al., 2024; Wieder et al., 2013). Recent efforts to improve model structures and develop new processes have emerged in some model groups, such as in ORCHIDEE-MICT (ORganizing Carbon and Hydrology in Dynamic EcosystEms–aMeliorated Interactions between Carbon and Temperature) – a version of the land surface component of the IPSL global climate model focused on resolving high latitude processes (Guimberteau et al., 2018). Key developments for SOC in this model include the representation of vertically-resolved organic carbon inputs from litter and depth-dependent decomposition processes to simulate the vertical discretization of soil carbon (Koven et al., 2009); the incorporation of effects of SOC on soil thermal properties to improve the simulated SOC in the top meter (Koven et al., 2009); the representation of vertical mixing of SOC via cryoturbation and bioturbation to vertically distribute SOC to a depth of ~3 m (Koven et al., 2009); the representation of sedimentation processes to simulate SOC below 3 m over Yedoma deposits (Zhu et al., 2016); the incorporation of peat-specific hydrology and carbon accumulation processes to simulate the dynamic peat carbon accumulation (Qiu et al., 2018, 2019). However, these advancements, developed in separate branches of ORCHIDEE-MICT, have yet to be integrated together for a realistic, process-based simulation of vertical and horizontal SOC distributions across the northern high latitudes.

Taking ORCHIDEE-MICT (hereafter MICT) as a starting point, in this study we present a new protocol for the accumulation of both shallow and deep soil organic carbon in the northern high latitudes (30°N-90°N) by: 1) merging the developed processes for SOC from various branches of MICT, including the carbon accumulation for non-peat vegetation types (Guimberteau et al., 2018), peat carbon accumulation (Qiu et al., 2018, 2019), sedimentation processes for Yedoma deposits (Zhu et al., 2016), and tile-specific energy budgets (Xi et al., 2024); and 2) representing the observed history of peat carbon evolution by integrating peat inception time according to radiocarbon dates measured from peat cores (Hugelius et al., 2020; Loisel et al., 2017). The simulated results with the new protocol are compared with those simulated with the old carbon accumulation protocol, to evaluate if the new protocol better reproduces observed SOC distributions.

## 2 Data and methods

To implement the model development, we used a branch version of ORCHIDEE-MICT, namely ORCHIDEE-MICT-teb (r8205), from Xi et al. (2024) as our base model. This version has PFT-specific (plant functional type-specific) energy budgets, soil thermics, and their interactions with carbon and hydrological processes, and has been comprehensively evaluated for its performance in simulating carbon, energy, and hydrological processes (Xi et al., 2024). To simulate Yedoma carbon accumulation, we incorporated sedimentation processes from Zhu et al. (2016), adapting them to be compatible with the PFT-specific framework of the new model. The model version used by Zhu et al. (2016) relied on grid-cell-averaged energy budgets, which meant that Yedoma carbon was mixed with coexisting PFTs (bare soil, tree, grass, crop) to regulate the average energy budget. Given that Yedoma soils typically have much higher carbon densities than general soils, accounting for their distinct carbon-energy-water feedbacks such as the thermal insulation effect of SOC (Koven et al., 2009; Zhu et al., 2019) is essential for realistically simulating both their historical development and future evolution. Regarding peatland processes, the base version already included peatland carbon accumulation. We thus did not develop new peatland processes, but we revised the way peatland initiation is simulated during the Holocene. Instead of assuming a homogeneous peatland age across northern peatlands starting in early Holocene, we employed spatially explicit peatland age maps based on observational data to define the onset of peat development at each grid cell. This was achieved by prescribing peatland cover maps at different epochs during the Holocene, which the model uses as a forcing to simulate realistic peatland expansion over time. As a result, the improved model allows for the independent simulation of Yedoma carbon, peatland carbon, and conventional soil carbon dynamics within a single grid cell.

In Section 2.1, we describe the soil carbon models in detail, including the general soil carbon processes and the peatland carbon accumulation scheme in the base version (Xi et al., 2024), as well as the Yedoma carbon accumulation processes from Zhu et al. (2016). Section 2.2 then presents the setup of the carbon accumulation simulations, using both the base version (without Yedoma carbon accumulation and with uniform peatland inception timing) and the improved version (with Yedoma carbon accumulation and spatially varying peatland inception).

### 2.1 Soil carbon model description

The soil carbon model in the original MICT (v8.4.1) (Guimberteau et al., 2018) is based on the CENTURY model (Parton et al., 1988), with two litter carbon pools (metabolic and structural) and three soil carbon pools (active, slow, and passive) defined by their turnover rates. The turnover time at 5 °C without moisture limitation for the five carbon pools is 0.37, 1.4, 0.84, 31, and 1,363 years, respectively. The vertical transfer and accumulation of soil carbon are driven by three main processes: root-density-dependent organic carbon inputs, depth-dependent decomposition of soil carbon regulated by vertically-layered soil temperature and soil moisture (considering pore ice), as well as carbon diffusion via bioturbation by animal and plant activity and via cryoturbation in permafrost soils. The assumed soil depth for soil hydrology is set to 2 m (11 layers) and the total depth

for soil thermal / carbon processes is 38 m (32 layers). The hydrological and thermal / carbon processes share the first 11 layers, while the thermal layers below use the soil moisture from the 11[th] hydrology layer. Limited by the root depth and active layer thickness, however, the deepest soil carbon simulated by the model is ~3 m. Cryoturbation can bury carbon below the active layer in the model, but this buried carbon merely exceeds a depth of 3 m. More details on soil carbon processes in MICT can be found in Krinner et al., (2005), Koven et al., (2009), Zhu et al., (2016), Guimberteau et al., (2018), and Huang et al., (2018).

Qiu et al. (2018; 2019) introduced a new soil tile to host peatland into MICT. Due to the unique highly-saturated, oxygen-deficient, and nutrient-limited soil environments in peatlands, this soil tile is covered by specific vegetation - a C3 grass plant functional type (PFT) with a shallower rooting depth, and is prescribed with specific soil hydrology properties, including hydraulic conductivity and diffusivity. Further, the peat soil tile has no drainage flux at the base of soil and accepts lateral surface water input from other biomes within the same grid cell (Qiu et al., 2018). The water-logged soil greatly reduces decomposition rates and leads to the accumulation of deep organic matter in peatlands. Instead of modeling the upward accumulation of peat, Qiu et al. (2019) simulated a downward sedimentation of peat carbon in the model, inspired by Jafarov and Schaefer, (2016). When the simulated SOC at soil layer $l$ is larger than a prescribed fraction (0.95) of observation-based carbon density ($C_{obs,l}$, kg m$^{-2}$), a prescribed fraction (0.05) of SOC will be moved down to the layer below ($l + 1$), resulting in accumulation of carbon in peat soils over time. The $C_{obs,l}$ threshold is calculated as

$$C_{obs,l} = BD_l \times \alpha_{c,l} \times \Delta Z_l \qquad (1)$$

where $BD_l$ (kg m$^{-3}$) is the soil bulk density at model layer $l$ (estimated with observation data), $\alpha_{c,l}$ (wt %) is the mass fraction of carbon in the soil (derived from observation data), $\Delta Z_l$ (m) is the thickness of soil layer $l$. Compared with northern peatland sites and gridded simulations over the Northern Hemisphere (> 30°N), the model generally reproduces the measured SOC density, peat depth from peat cores, and the total northern peatland area (3.9 Mkm$^2$ against 3.4-4.0 Mkm$^2$), total northern peat carbon stocks (463 Pg C against 270-540 Pg C) from previous studies (Qiu et al., 2019).

The simulation of the sedimentation processes forming Yedoma deposits was introduced by Zhu et al. (2016). Similar to peat carbon accumulation, the basic idea for modeling the upward accumulation of carbon when Yedoma deposits form is to simulate an equivalent downward transport of organic carbon (OC) in the model, by employing a vertical advection equation:

$$\frac{\partial C_i(z,t)}{\partial t} + u(t)\frac{\partial C_i(z,t)}{\partial z} = f_i(z,t) - g_i(z,t) \times C_i(z,t) \qquad (2)$$

where $C_i(z,t)$ (g m$^{-3}$) is OC concentration of litter or soil carbon pool $i$ at depth $z$ and time $t$; $u(t)$ (m d$^{-1}$) is the downward rate of OC; $f_i(z,t)$ (g m$^{-3}$ d$^{-1}$) is the OC input to pool $i$; and $g_i(z,t)$ is the decomposition rate of pool $i$. The downward rate of OC, i.e., the sedimentation rate, can be inferred from the height-age slope of Yedoma deposits based on site measurements. Since the height of Yedoma deposits is at landscape scale, the inferred sedimentation rate includes ice wedge (~50% in volume). In this way, although the representation of ice wedges is absent in MICT, the volume of ice wedges is implicitly included in the

model. To reach a mean deposit thickness of 19.4 m suggested by (Strauss et al., 2013), four typical sedimentation rates (0.6, 0.8, 1.0, and 1.2 mm yr$^{-1}$) observed in Yedoma sites with corresponding accumulation duration (32, 24, 19, and 16 thousand years, hereafter kyr) were assessed to simulate present-day Yedoma deposits. The simulated soil carbon stock (125-145 Pg C) in present-day Yedoma region (0.42 Mkm$^2$) by the model is comparable to the estimate by Strauss et al. (2013) (83 + 61/-57 Pg C).

## 2.2 Simulation setup

Conventional LSMs employ a 'spin-up' simulation to equilibrate the biogeochemical cycles before calculating perturbations. A perfect spin-up with a perfect model would accumulate SOC stocks to match exactly the observations. For instance, in the TRENDY (Trends in the land carbon cycle) project, the spin-up simulation is run with recycled climate from the first two decades of the 20th century (1901-1920), atmospheric $CO_2$ concentration from the year 1700 (276.59 ppm), and a land cover map fixed to the year 1700 (Sitch et al., 2015). This classical spin-up procedure is a strong simplification, since in reality, soils were not in a steady state under pre-industrial climatic conditions. To address this problem, we designed a new spin-up protocol, aiming at reproducing the formation of deep carbon in Yedoma deposits and peatland soils since the late Pleistocene (see a schematic representation in Fig. 1). Below, we describe both the classical (or old) spin-up simulation with the base version (Xi et al., 2024), as used in the TRENDY project, and the new spin-up protocol with the improved MICT version which includes the representation of deep carbon accumulation in Yedoma deposits and observed history of peat carbon evolution.

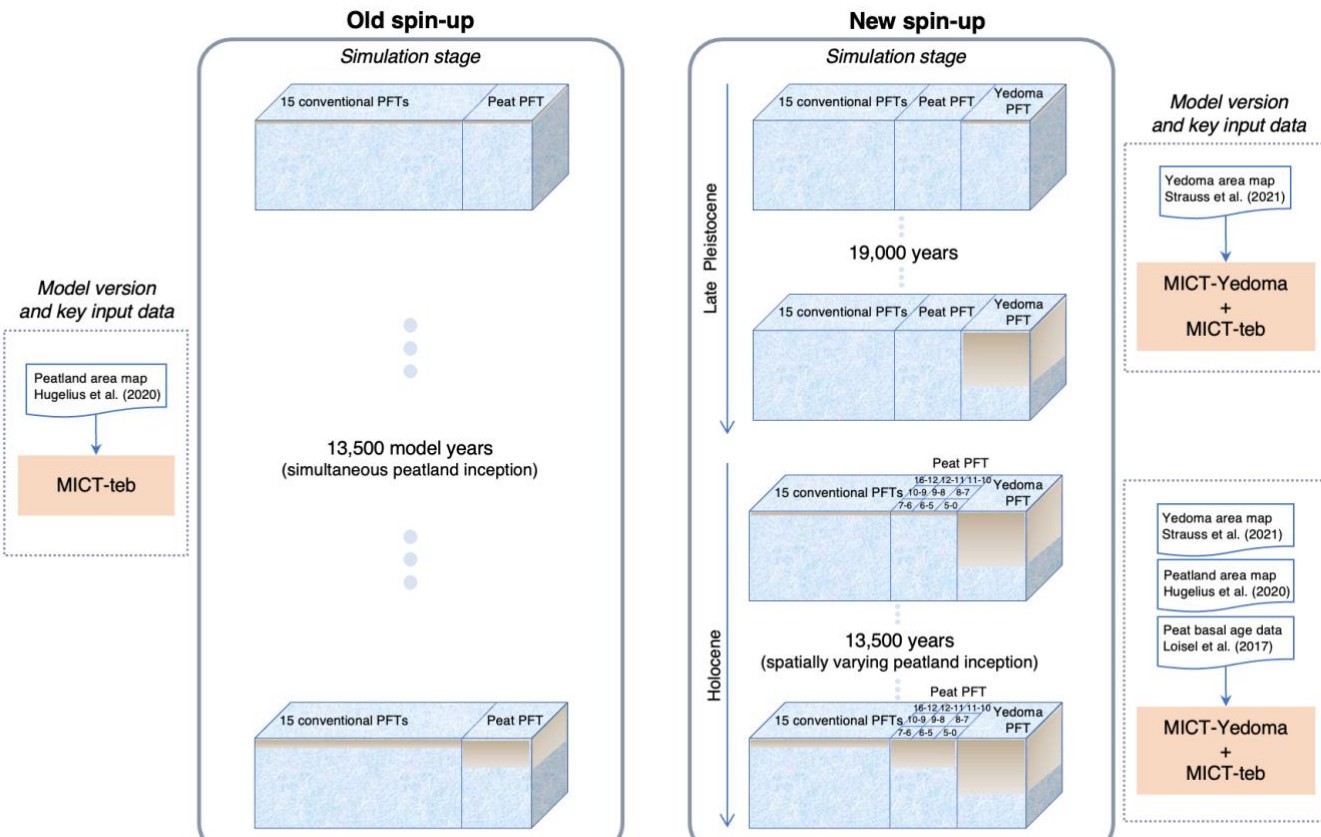

**Figure 1: Schematic representation of soil carbon accumulation within one grid cell across different simulation stages in the old and new spin-up simulations. The old spin-up uses the ORCHIDEE-MICT-teb (MICT-teb) model version, includes 16 plant functional types: 15 conventional PFTs (bare soil, 8 tree PFTs, 4 grass PFTs, and 2 crop PFTs) and one peatland PFT (C3 grass) within a grid cell (Table S1). In the old spin-up, the cover fraction of peatland is fixed throughout the simulation, and all peatlands are assumed to have formed simultaneously, 13,500 years ago. By contrast, the new spin-up merges ORCHIDEE-MICT-Yedoma (MICT-Yedoma) with MICT-teb, enabling the model to simulate Yedoma carbon accumulation during the late Pleistocene and subsequent carbon accumulation in peatland and conventional soils during the Holocene. The new spin-up includes the same 15 conventional PFTs, one peatland PFT, and an additional Yedoma PFT (C3 grass) within a grid cell (Table S1). Peatland cover varies dynamically across nine Holocene epochs (shown in the peat PFT domain), based on prescribed maps constructed from peat basal age observations and present-day peatland distribution.**

### 2.2.1 Old spin-up simulation

Table 1 shows the protocol of the old spin-up simulation for carbon accumulation with the starting MICT version. It includes three stages: 1) 100 years with the full ORCHIDEE land surface model (FullO) run to initiate the energy cycle, hydrological processes, and vegetation growth in the model (the upper grid cell of the old spin-up in Fig. 1); 2) 13,500 years with a stand-alone soil carbon module (SubC) run to accelerate carbon accumulation (using fixed monthly litter input, soil moisture, and soil thermal conditions from the preceding FullO simulation); and 3) 100 years with the FullO run to bring energy, hydrology,

and carbon back into equilibrium (the lower grid cell of the old spin-up in Fig. 1). All three stages of the old spin-up simulation use the same forcing datasets, including climate data recycling 1901-1920 from CRU-JRA v2.3 (the version used in Global Carbon Budget 2021) (Friedlingstein et al., 2022), the fixed $CO_2$ level in 1700 (276.59 ppm), and the fixed land cover map generated by combining the 1700 land cover map from TRENDY for 15 conventional PFTs (bare soil, 8 tree PFTs, 4 grass PFTs, and 2 crop PFTs; see more details in Table S1) (Lurton et al., 2020) and the peat map from Hugelius et al. (2020) for the peat C3 grass PFT. All the simulations were run for the Northern Hemisphere (30°N-90°N) at a 2° × 2° spatial resolution. Thus, the old spin-up simulation simply runs the model for a very long period (13,500 years) to accumulate enough soil carbon. However, it doesn't include Yedoma deposits, and although the starting MICT version used in the old spin-up has merged the peat carbon accumulation, it uses a uniform duration (13,500 years) of peat carbon accumulation for all peatlands and therefore misses the evolution of peatland extent as in reality (Fig. 1).

**Table 1: Protocol for the old spin-up simulation. FullO indicates the full ORCHIDEE-MICT and SubC indicates the stand-alone soil carbon module.**

| Simulation | Period | Climate | CO₂ | Land cover map |
|---|---|---|---|---|
| FullO | 100 yr | | | |
| SubC | 13,500 yr | CRUJRA 1901-1920 cycling | 276.59 ppm | TRENDY 1700 + peat incepted before 1-0 ka |
| FullO | 100 yr | | | |

### 2.2.2 New spin-up simulation

Since radiocarbon dating of Yedoma deposits and peat cores provides their age of deposition, we are no longer entirely blind to the history of soil carbon evolution. In the new spin-up simulation, we explicitly represented the timing of inception and the duration of accumulation for Yedoma deposits and peat carbon (Fig. 1). Table 2 summarizes the new spin-up simulation.

- **Yedoma deposits**

In the real world, Yedoma deposits accumulated over tens of thousands of years due to the cold climatic and environmental conditions in the late Pleistocene glacial period, with about 70% of them undergoing thermokarst degradation or fluvial erosion during the warmer Holocene (Grosse et al., 2013; Strauss et al., 2017). In the new spin-up, after merging the Yedoma sedimentation processes developed by Zhu et al. (2016) into the starting MICT version, we used a 1 mm yr$^{-1}$ sedimentation rate and a 19 kyr accumulation period as in their study to simulate carbon stocks in Yedoma deposits under the Last Glacial Maximum (LGM) climate (21-18 thousand years before present, hereafter ka). A map of present-day Yedoma from Strauss et al., (2021), covering 0.48 Mkm$^2$ of Yedoma deposits (Fig. 2), was used to limit the model to predict Yedoma deposits which

have survived the deglaciation. This was done because we aim to represent Yedoma carbon during the pre-industrial period in this study. We didn't test other sedimentation rates and accumulation periods here because Zhu et al. (2016) found that it has a small impact on simulated total SOC stock. Other deposits, such as those from thermokarst lake basins, were not included in this study because they were deposited via different processes. To accelerate the carbon accumulation, we used the "FullO-SubC-FullO" strategy (Table 1) in the 19 kyr simulation. We used the LGM climate forcing data output from the Institut Pierre

Simon Laplace (IPSL) earth system model, IPSL-CM5A-LR (Kageyama et al., 2013), the LGM $CO_2$ level at 185 ppm, and the LGM vegetation growth simulated by the dynamic vegetation model in MICT due to the lack of land cover maps during that period (Zhu et al., 2015). Consistent with the old spin-up simulation, the Yedoma simulation was also run at a $2° \times 2°$ grid resolution, but only for 60°N-90°N to save computation cost, given the relatively smaller spatial extent of Yedoma deposits (Fig. 2; Strauss et al., 2021). The schematic representation of Yedoma carbon accumulation is shown with the two upper grid

cells for the new spin-up in Fig. 1. Further details on how the Yedoma simulation is connected with the peatland simulation are provided in the following subsection.

Table 2: Protocol for the new spin-up simulation. FullO indicates the full ORCHIDEE-MICT land surface model and SubC indicates the stand-alone soil carbon module.

| Yedoma simulation | | | | | |
|---|---|---|---|---|---|
| **Epoch** | **Simulation** | **Period** | **Climate** | **CO₂** | **Land cover map** |
| 21-18 ka | FullO | 100 yr | 100-yr LGM climate cycling | 185 ppm | Simulated by dynamic vegetation model in MICT |
| | SubC | 19,000 yr | | | |
| | FullO | 100 yr | | | |
| **Peat simulation** | | | | | |
| **Epoch** | **Simulation** | **Period** | **Climate** | **CO₂** | **Land cover map** |
| 13.5-11.5 ka | FullO | 100 yr | CRUJRA 1901-1920 cycling | 238 ppm | TRENDY 1700 + peat incepted during 16-12 ka + Yedoma PFT |
| | SubC | 2,000 yr | | | |
| 11.5-10.5 ka | FullO | 100 yr | CRUJRA 1901-1920 cycling | 250 ppm | TRENDY 1700 + peat incepted during and before 12-11 ka + Yedoma PFT |
| | SubC | 1,000 yr | | | |
| 10.5-9.5 ka | FullO | 100 yr | CRUJRA 1901-1920 cycling | 265 ppm | TRENDY 1700 + peat incepted during and before 11-10 ka + Yedoma PFT |
| | SubC | 1,000 yr | | | |
| 9.5-8.5 ka | FullO | 100 yr | CRUJRA 1901-1920 cycling | 265 ppm | TRENDY 1700 + peat incepted during and before 10-9 ka + Yedoma PFT |
| | SubC | 1,000 yr | | | |

| | | | | | |
|---|---|---|---|---|---|
| 8.5-7.5 ka | FullO | 100 yr | CRUJRA 1901-1920 cycling | 260 ppm | TRENDY 1700 + peat incepted during and before 9-8 ka + Yedoma PFT |
| | SubC | 1,000 yr | | | |
| 7.5-6.5 ka | FullO | 100 yr | CRUJRA 1901-1920 cycling | 260 ppm | TRENDY 1700 + peat incepted during and before 8-7 ka + Yedoma PFT |
| | SubC | 1,000 yr | | | |
| 6.5-5.5 ka | FullO | 100 yr | CRUJRA 1901-1920 cycling | 262 ppm | TRENDY 1700 + peat incepted during and before 7-6 ka + Yedoma PFT |
| | SubC | 1,000 yr | | | |
| 5.5-4 ka | FullO | 100 yr | CRUJRA 1901-1920 cycling | 265 ppm | TRENDY 1700 + peat incepted during and before 6-5 ka + Yedoma PFT |
| | SubC | 1,500 yr | | | |
| 4-0 ka | FullO | 100 yr | CRUJRA 1901-1920 cycling | 272 ppm | TRENDY 1700 + peat incepted during and before 5-0 ka + Yedoma PFT |
| | SubC | 4,000 yr | | | |
| | FullO | 100 yr | CRUJRA 1901-1920 cycling | 276.59 ppm | TRENDY 1700 + peat incepted during and before 5-0 ka + Yedoma PFT |

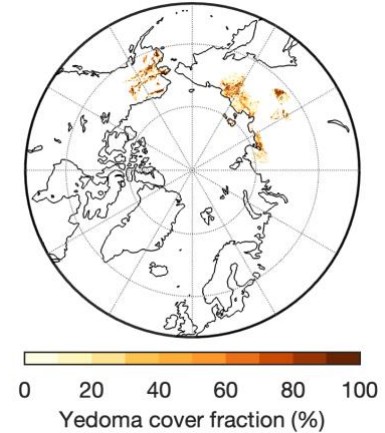

**Figure 2: Spatial distribution of present-day Yedoma region from Strauss et al. (2021), shown at the initial spatial resolution of 0.05° × 0.05°. The spatial extent of Yedoma in the map covers only areas north of 60°N.**

● **Peatlands**

The radiocarbon dates obtained from peat cores have revealed the heterogeneous inception times of global peatlands (MacDonald et al., 2006; Loisel et al., 2017), suggesting that it's unreasonable to treat all peatlands with a uniform duration (13,500 years) of carbon accumulation as in the old spin-up (Treat et al., 2019, 2021). To address this, we used peat inception maps created according to radiocarbon dates from 2,860 geolocated peat cores (Fig. S1) presented in Loisel et al. (2017). These

maps summarize the mean peat inception ages grouped by 1°×1° grid cells. The data from these maps are prescribed into the model to represent the observed history of peat carbon evolution. The earliest timing of peatland inception according to these peat cores was around 16 ka. During that period, deglacial warming halted Yedoma accumulation, causing the thaw of massive ice wedges in many Yedoma deposits, which resulted in ground subsidence and the formation of thermokarst lakes (Anthony et al., 2014; Brosius et al., 2021; Strauss et al., 2017). Where thaw created drainage channels, subsequent lake drainage lowered water levels, enabling the growth of benthic mosses and other plants whose accumulation would ultimately lead to peatland formation (Anthony et al., 2014). Accordingly, in the new spin-up simulation, we disabled Yedoma sedimentation over present-day Yedoma regions after the LGM and initiated peat carbon accumulation in suitable areas where peat inception was observed. The suitable areas and observed inception time was determined by combining the point-level peat age data (Fig. S1) and our peatland fraction map (Hugelius et al., 2020) and transformed into gridded peatland cohort maps covering 30°N-90°N for each thousand-year interval since 16 ka (Fig. S2).

Based on the gridded peatland cohort maps we computed the number of peatland grid cells and total peatland area at different epochs of the pre-Holocene deglaciation and Holocene period from 16 ka to present (Fig. 3). Since that few peatlands formed during the earliest period (16-12 ka) and the most recent one (5-0 ka), compared to each kyr from 12 ka to 5 ka, we used the area-weighted peatland inception time as the inception time of peatlands during these two epochs and reduced the simulation time in kyrs from 16 to 9 to save computational cost. The peatland inception time of the 9 epochs or cohorts are 13.5, 11.5, 10.5, 9.5, 8.5, 7.5, 6.5, 5.5, and 4 ka, respectively (Table 2). We also used the "FullO-SubC" strategy to accelerate the carbon accumulation in the peat profiles. For each kyr, we used the 20-yr cycled climate data from CRUJRA v2.3 (1901-1920) as in the old spin-up simulation, the $CO_2$ levels for each epoch derived from the ice-core record (obtained from https://www1.ncdc.noaa.gov/pub/data/paleo/icecore/antarctica/domec/), and the land cover maps combining 15 conventional PFTs using the 1700 land cover map from TRENDY, one peat PFT (C3 grass) defined by the gridded peatland cohort maps during each kyr (Fig. S2), and one Yedoma PFT (C3 grass) defined by the Yedoma map from Strauss et al. (2021). We reintroduced Yedoma during this phase because our study is aimed to simulate Yedoma deposits that still exist today, that is, deposits that survived through the Holocene. To do this, we transferred the litter and soil carbon from the final year of the Yedoma simulation to initialize the carbon pools of the Yedoma PFT in this phase. However, we turned off sedimentation, letting the Yedoma PFT follow the same processes as conventional PFTs and accumulate carbon slowly. The fraction of the Yedoma PFT during each kyr of peat inception was fixed since this Yedoma was assumed to have survived until today. When creating the land cover maps, we prioritized Yedoma and peat fractions in each grid cell and adjusted the fractions of 15 conventional PFTs from the TRENDY land cover map proportionally. As a result, compared to the old spin-up simulation, our new spin-up simulation included carbon accumulation for Yedoma deposits and the 'realistic inception history' for peatland evolution from 16 ka to present (Fig. 1). All the simulations for this phase were run for 30°N-90°N at a 2° × 2° spatial resolution, consistent with the old spin-up simulation.

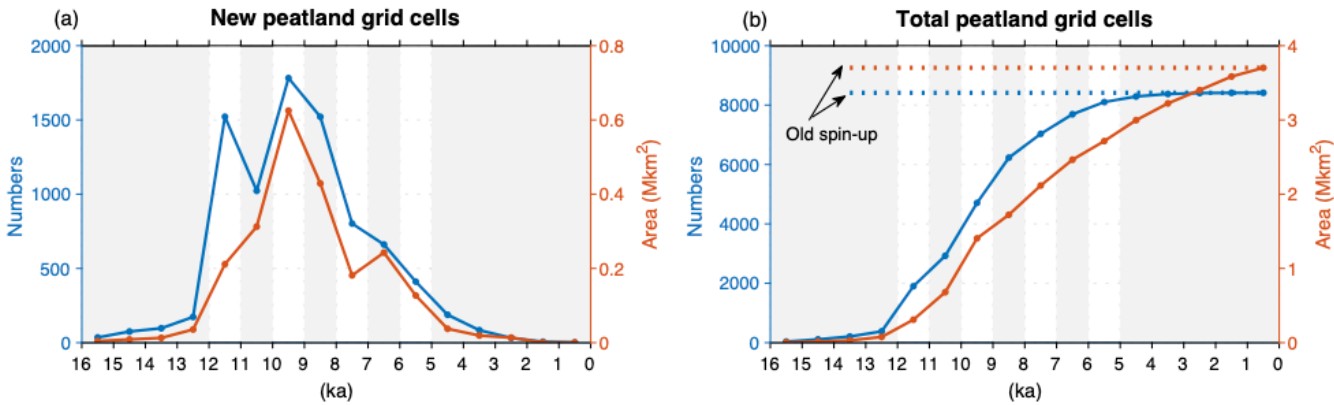

**Figure 3: Evolution of northern peatlands (> 30°N) from 16 ka to present according to gridded peatland cohort maps as shown in Fig. S2. (a) Number and area of new peatland grid cells (with the initial spatial resolution of 1° × 1°) incepted during each thousand years from 16 ka to present. (b) Number and area of total peatland grid cells by each thousand years from 16 ka to present. Peatland grid cells are defined as grid cells with a > 0 peatland fraction. In Fig. (b), the numbers and area of total peatland grid cells used in the old spin-up are also shown, which misses the evolution of peatland extent from 13.5 ka to present.**

- **Conventional soils**

There's no new setup for conventional soils, where the 15 conventional PFTs (Table S1) grow, in the new spin-up simulation, compared to the old spin-up. However, it's noteworthy that in the starting MICT version from Xi et al., (2024), the SOC simulated by the model can feed back to soil temperature–an important mechanism for simulating the formation of high-latitude SOC stocks (Koven et al., 2009), while in the original MICT version, Guimberteau et al., (2018) used the observation-based SOC maps to represent the insulation effect of SOC on soil thermal diffusivity.

- **Changes in plant and soil carbon pools following conversion from conventional soils to peatlands**

In the starting MICT version from Xi et al., (2024), when conventional soils are converted to peatlands within a grid cell, the newly established peatland fraction initially inherits the plant and soil carbon pools from the displaced conventional soils to ensure mass conservation. After this transition, the peatland fraction begins to grow peatland-specific PFT and accumulate soil carbon according to peatland soil characteristics. As described in Section 2.1, compared to conventional PFTs (e.g., forest or grass), the peatland PFT features a shallower rooting depth and is assigned distinct hydrological properties. Moreover, the peat soil tile does not allow drainage at the base of the soils and also receives lateral surface water input from other non-peatland PFTs within the same grid cell. These water-logged soil conditions substantially suppress decomposition, promoting the accumulation of soil organic carbon. Peatland tiles also differ regarding vertical soil carbon transfer. Conventional PFTs can accumulate carbon down to a depth of 3 m through root-distributed inputs, depth-dependent decomposition, and vertical diffusion via bioturbation and cryoturbation (see the first paragraph of Section 2.1). In contrast, the peatland PFT employs a more efficient scheme for vertical carbon transfer (detailed in the second paragraph of Section 2.1), enabling substantial carbon accumulation down to depths of up to 10 m.

## 3 Results and evaluation

In this section, we present the simulated Yedoma carbon accumulation from the improved model version in Section 3.1. In Section 3.2, we examine how peatland soil carbon accumulation responds to the implementation of spatially varying peatland initiation timing. Since the vertical carbon transfer process in Yedoma is driven by deposition rates derived from site-level measurements, and peatland development is informed by prescribed maps extrapolated from point-based peat age data, we evaluate model performance by comparing simulated soil carbon from both the old and new spin-up schemes, as well as against site-level observations. In Section 3.3, we provide a spatial evaluation using the deepest available gridded soil carbon map (to our knowledge), although it only covers the top 0-3 m of soil.

### 3.1 Yedoma carbon accumulation

Figure 3 shows the Yedoma carbon accumulation from the new spin-up simulation. With a 1 mm yr$^{-1}$ sedimentation rate and a 19 kyr accumulation duration under LGM climate, the model accumulates ~20 m of organic carbon on average over present-day Yedoma region (Fig. 4b). This is very close to the mean deposit thickness of 19.4 m suggested by Strauss et al., (2013). The mean simulated carbon density over the soil column was 271 kg C m$^{-2}$, with a large spatial variation given an interquartile range of 366 kg C m$^{-2}$ (Fig. 4a). Vertically, the Yedoma SOC tended to be near-evenly distributed along soil layers, with a profile-mean volumetric carbon density of ~14 kg C m$^{-3}$ (Fig. 4c). The spatial patterning and vertical distribution of Yedoma SOC is consistent with the outputs from Zhu et al. (2016), and the total SOC stock from the new spin-up (141 Pg C) is within the range of their estimate (125-145 Pg C). When excluding the volume of ice-wedges (~50% of soil column volume) used to compute the sedimentation rate in Eq. (2), the simulated SOC density from our model (~28 kg C m$^{-3}$) compares well with measurements, e.g., $25.98 \pm 1.5$ kg C m$^{-3}$ (excluding ice) in Anthony et al., (2014), and 15.4-26.8 kg C m$^{-3}$ (excluding ice) in Palmtag et al., (2022). Given the significant uncertainties in the Yedoma area and the scarce measurements of deep Yedoma SOC profiles, these results suggest that the new spin-up reasonably reproduces the Yedoma carbon stock.

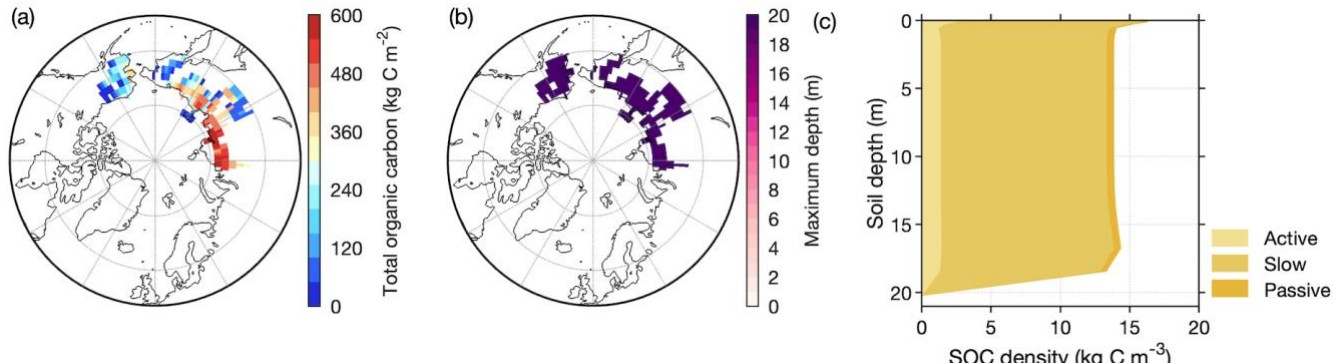

**Figure 4: Yedoma carbon accumulation from the new spin-up simulation. Spatial distribution of simulated (a) total organic carbon (per m$^2$ Yedoma deposits) and (b) the maximum depth of total organic carbon, and (c) the simulated mean vertical distribution of carbon density of three soil carbon pools (active, slow and passive) over present-day Yedoma region by the end of the LGM. The**

spatial extent of Yedoma coverage data in maps (a) and (b) are limited to areas north of 60°N, due to the smaller spatial extent of Yedoma (Fig. 2) compared to our entire study domain (30°N-90°N).

## 3.2 Carbon accumulation in peat and other soils after the LGM

Regarding carbon accumulation in peat and other soils, in the old spin-up simulation, the total organic carbon (TOC, including litter and soil organic carbon) for peat and 15 conventional PFTs over the Northern Hemisphere (30°N-90°N) gradually accumulates from 0 to ~287 Pg C during the first simulation stage (Fig. 5c), driven by a large positive net ecosystem productivity (NEP) resulting from the excess of net primary productivity (NPP) over heterotrophic respiration (Rh) (Fig. 5a

350 and Fig. S3). Subsequently, the long (13,500 years) integration of the stand-alone soil carbon module resulted in a TOC increase of ~1,500 Pg C, followed by a nearly-stabilization period (NEP = 0.6 Pg C yr$^{-1}$, of which 0.4 for conventional PFTs and 0.09 for peat PFT) during the third simulation stage.

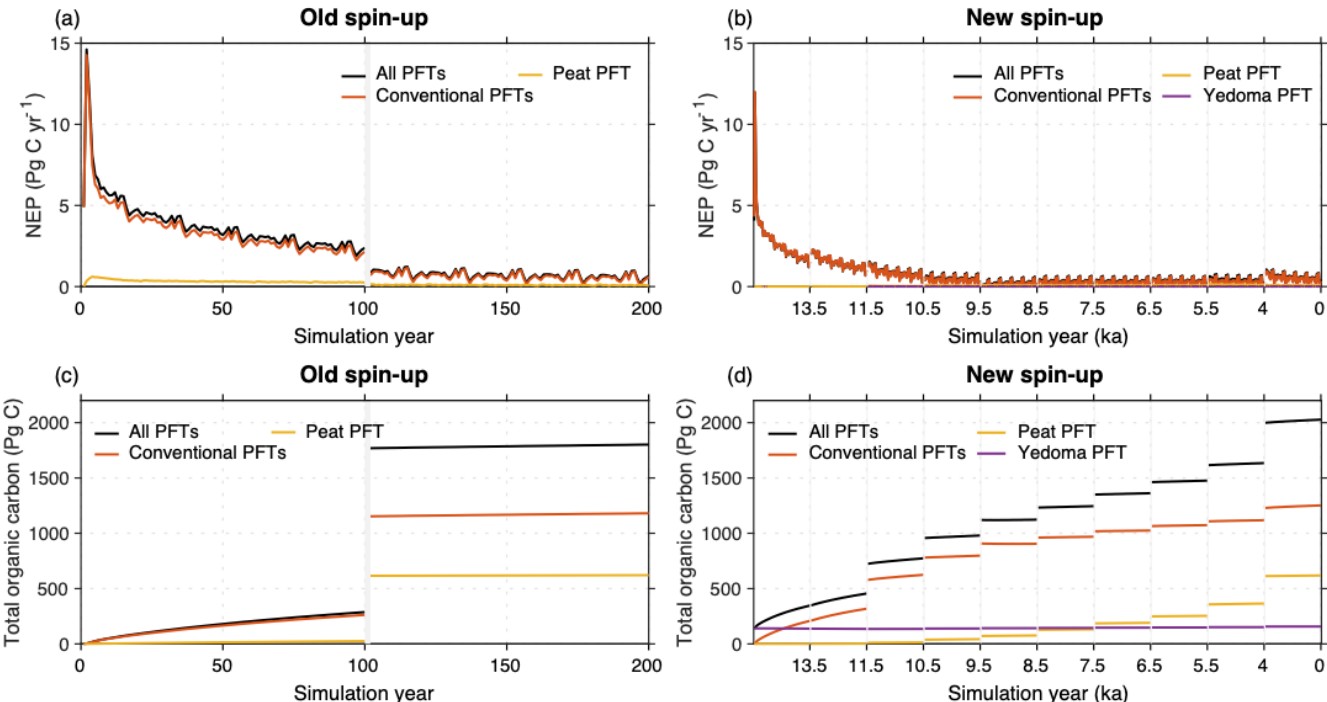

**Figure 5: Time series of net ecosystem productivity (NEP) and total organic carbon over the Northern Hemisphere (> 30°N) from the old ((a) and (c)) and the new ((b) and (d)) spin-up simulations. Conventional PFTs indicate the 15 non-yedoma and non-peat PFTs including bare soil, 8 tree PFTs, 4 grass PFTs, and 2 crop PFTs. The simulation periods using the stand-alone soil carbon module (SubC) are not plotted in all figures as the model only outputs the simulation results of the last year of each period to save storage space.**

With its explicit representation of peatland inception times, the new spin-up simulation shows a similar temporal trend in NEP from 16 ka to present as the old spin-up (Fig. 5b). By the end of the simulation, annual NEP is a carbon sink of 0.4 Pg C yr$^{-1}$ (0.3 for conventional PFTs, 0.08 for peat PFT, and 0.007 for Yedoma PFT). Accordingly, the conventional PFTs and peat PFT accumulate 1,239 Pg C and 632 Pg C, respectively, while the Yedoma PFT holds a mean carbon stock of 144 Pg C throughout the entire simulation period (Table 3). Since we only simulated sedimentation processes over present-day Yedoma, it's reasonable that the simulated Yedoma carbon stocks deposited during the LGM climate could persist during the warm deglacial period. Due to increased vegetation productivity under the warmer Holocene climate compared to the LGM, the simulated Yedoma carbon stock increases by 16 Pg C through the Holocene, which matches well with the estimates of 13 ± 1 Pg C from Strauss et al., (2017). For the peat PFT, we found that the peatland carbon accumulation rate at the end of the simulation from the new spin-up (0.08 Pg C yr$^{-1}$) was quite close to that from the old-spin up (0.09 Pg C yr$^{-1}$), both of which are comparable with the observed rate (0.10 (0.02-0.19) Pg C yr$^{-1}$) (Li et al., 2023). The TOC for peat PFT from the two spin-ups is almost identical (Table 3). For the 15 conventional PFTs, the inclusion of Yedoma PFT in the land cover map slightly reduced their area (0.7%), while TOC is 68 Pg C (9%) higher in the new spin-up simulation (Table 3). As a result, the TOC over the Northern Hemisphere (> 30°N) for all PFTs is 2,028 Pg C in the new spin-up, 226 Pg C more than the old spin-up.

To better understand the difference in TOC between the two spin-up simulations, we computed carbon stocks across four depth intervals (0-1 m, 1-2 m, 2-3 m, and 3-38 m), Arctic and Boreal regions (60-90°N and 30-60°N), and all the carbon pools (litter, active soil, slow soil, and passive soil) (Table 3). The litter OC for conventional PFTs is almost identical between the old and new spin-ups, but it is 4.2 Pg C (+22%) higher for peat PFT in the new spin-up. For soil carbon, both conventional PFTs and peat PFT show higher SOC for 0-2 m soil (82 Pg C (+8%) and 17 Pg C (+16%), respectively) but lower SOC for 2-38 m soils (13 Pg C (-24%) and 21 Pg C (-6%), respectively) in the new spin-up. The higher SOC for conventional PFTs is concentrated in the Arctic region (85 Pg C, +15%), predominantly in the slow SOC pool (101 Pg C, +19%). By contrast, the peat PFT also has higher SOC (49 Pg C, +15%) in the Arctic region but lower SOC (53 Pg C, -18%) in the Boreal region, which offsets and even reverses the SOC gains in the Arctic region in the new spin-up. Breaking down the PFT classes into their CENTURY-derived soil carbon quality pools, the peat PFT has a higher active SOC (34 Pg C, +75%) but a lower passive SOC (35 Pg C, -43%) in the new spin-up outputs. The lower simulated SOC for deep soils and in the Boreal region, as well as less passive SOC for the peat PFT could be attributable to the shorter duration of carbon accumulation after constraining peat inception time in the new spin-up. By comparing the peatland cells by age cohort, we found that those young peatland cells, e.g, incepted after 8-7 ka, in the new spin up show a 6-31% lower total SOC (Fig. 6a) and a 50-70% lower passive SOC (Fig. 6b-d), relative to the 13,500-year-old peatlands in the old spin-up. The higher SOC for 0-2 m soils in the high latitudes for both conventional PFTs and the peat PFT in the new spin-up could be associated with the setup of the new spin-up simulation and some existing model processes, which are discussed in the Discussion section.

**Table 3: Summary of area and organic carbon (OC) stocks for different PFT categories by the end of the old and new spin-up simulations. The OC stocks for four PFT groups (all PFTs, conventional PFTs, peat PFT, and Yedoma PFT) are shown by four depth intervals (0-1 m, 1-2 m, 2-3 m, and 3-38 m), two regions (60-90°N and 30-60°N), and four carbon pools (litter, active soil, slow soil, and passive soil). The combined observation-based product (WISE30sec for global, non-permafrost, 0-2 m soil layers and NCSCD for permafrost, 0-3 m soil layers) is also shown for comparison (see details in the Section 3.3).**

| PFT category | | | All PFTs | | | Conventional PFTs | | Peat PFT | | Yedoma PFT | |
|---|---|---|---|---|---|---|---|---|---|---|---|
| | | | WISE30sec +NCSCD | Old | New | Old | New | Old | New | Old | New |
| Area (Mkm²) | | | 68.6 | 68.6 | 68.6 | 65.0 | 64.5 | 3.6 | 3.6 | 0 | 0.5 |
| Total OC (Pg C) | | | 1663.3 | 1801.8 | 2027.5 | 1170.4 | 1238.6 | 631.4 | 632.0 | 0 | 157.0 |
| Litter OC (Pg C) | | | — | 86.5 | 91.1 | 67.3 | 66.4 | 19.2 | 23.4 | 0 | 1.4 |
| Soil OC (Pg C) | Total | | 1663.3 | 1715.4 | 1936.4 | 1103.1 | 1172.2 | 612.2 | 608.6 | 0 | 155.6 |
| | By depth | 0-1 m | 846.4 | 882.8 | 959.5 | 746.5 | 790.0 | 136.3 | 145.3 | 0 | 24.1 |
| | | 1-2 m | 524.8 | 427.2 | 490.9 | 303.1 | 341.5 | 124.1 | 131.9 | 0 | 17.4 |
| | | 2-3 m | 292.1 | 135.5 | 128.3 | 53.5 | 40.6 | 82.0 | 81.0 | 0 | 6.8 |
| | | 3-38 m | — | 269.8 | 357.6 | 0 | 0.03 | 269.8 | 250.3 | 0 | 107.3 |
| | By region | 60-90°N | 793.3 | 795.4 | 1084.9 | 473.3 | 557.8 | 322.1 | 371.5 | 0 | 155.6 |
| | | 30-60°N | 870.0 | 919.9 | 851.5 | 629.8 | 614.4 | 290.1 | 237.2 | 0 | 0 |
| | By pools | Active | — | 68.9 | 115.5 | 24.0 | 26.3 | 44.8 | 78.6 | 0 | 10.6 |
| | | Slow | — | 1016.6 | 1242.7 | 529.9 | 630.5 | 486.7 | 484.1 | 0 | 128.1 |
| | | Passive | — | 629.9 | 578.2 | 549.2 | 515.4 | 80.7 | 46.0 | 0 | 16.8 |

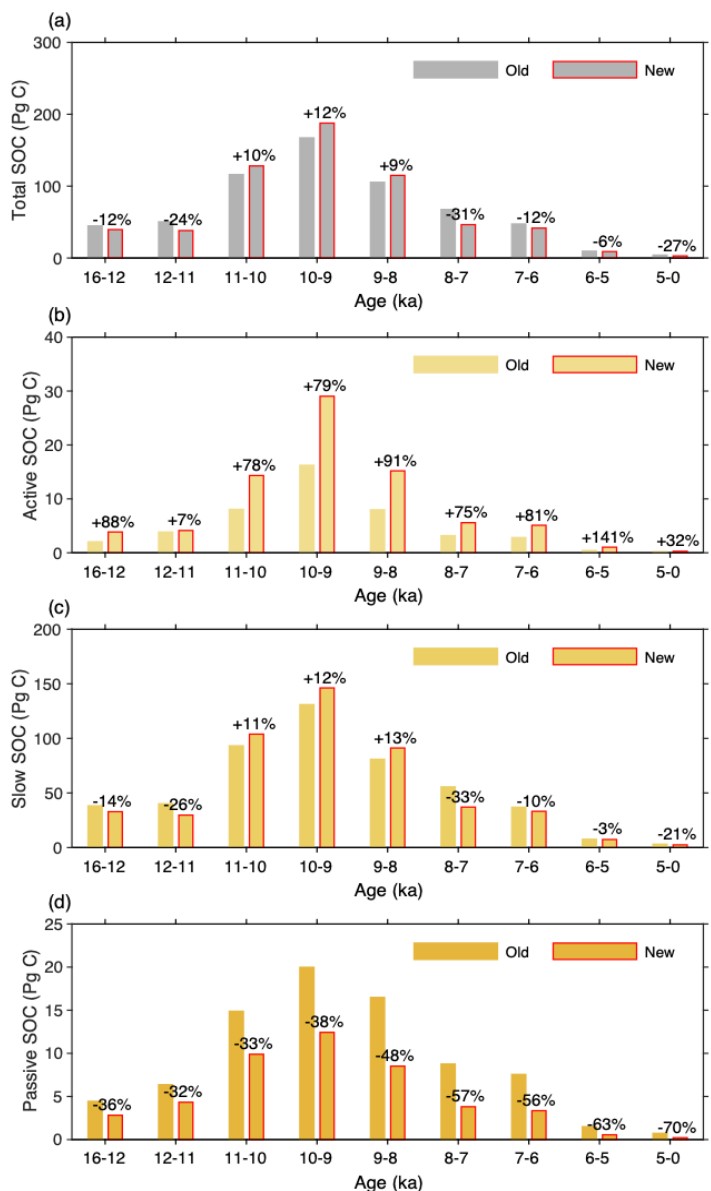

**Figure 6: Comparison of total (a), active (b), slow (c), and passive (d) SOC by age cohorts for peat PFT between the new and old spin-up simulations. The age cohorts are identified as the peatland inception time as shown in Fig. 3 and Table 2. The numbers show the relative difference between the new and old spin-ups.**

Figure 7 shows the spatial pattern of total organic carbon and the vertical profile of SOC density for three soil types after peat carbon accumulation simulation from the old and new spin-ups. Consistent with the results shown in Table 3, the higher total OC in the new spin-up is mainly distributed in the Arctic region (Fig. 7), where the inclusion of Yedoma carbon and the larger carbon stocks by conventional PFTs (Fig. S4) and peat PFT (Fig. S5) result in a > 50 kg C m$^{-2}$ higher carbon density (Fig. 7a-

c) and a > 5 m deeper maximum depth of SOC than the old spin-up (Fig. 7d-f). Analyzing the cumulative percentage of SOC at depths of 1 m, 2 m, and 3 m, we found that the higher SOC in very high latitudes (> 70°N) is mainly contributed by soil layers deeper than 3 m from Yedoma PFT (Fig. 7g), while the proportion of 0-3 m SOC in the new spin up is 10-20% lower than the old one (Fig. S6). In the Boreal region, the shorter duration of peat carbon accumulation in the new spin-up leads to a shallower SOC depth (1-5 m) and a lower carbon density (~10 kg C m$^2$) (Fig. 7h and Fig. S5). This shallower peat carbon results in a 5-25% increase in its share of SOC in the 0-3 m soil layers relative to the old spin-up in this region (Fig. S6).

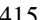

**Figure 7: Carbon accumulation for three soil types after the peat simulation from the old and new spin-ups. (a)-(c), Spatial distribution of total organic carbon over the Northern Hemisphere (> 30°N) from the old (a) and new (b) spin-up simulations, as**

**well as the difference between new and old spin-ups (c). (d)-(f), Spatial distribution of the maximum depth of soil organic carbon (SOC) from the old (d) and new (e) spin-up simulations, as well as the difference between new and old spin-ups (f). (g)-(i), Vertical distribution of carbon density of three soil carbon pools (active, slow and passive) for Yedoma PFT (g), peat PFT (h), and conventional PFTs (i) from the old and new spin-ups.**

## 3.3 Spatial evaluation against observation-based soil organic carbon maps

To further evaluate the soil carbon simulation with the new spin-up, we compared the spatial distribution of simulated total soil carbon with two observation-based soil datasets: WISE30sec (Batjes, 2016) and NCSCD (the Northern Circumpolar Soil Carbon Database) (Hugelius et al., 2013) (Fig. 8). WISE30sec provides global coverage of soil carbon stocks, but only for 0-2 m soil layers. In contrast, NCSCD is a regional dataset for permafrost regions but includes estimated soil carbon down to 3 m depth. We therefore combined the two datasets by using NCSCD for permafrost regions and WISE for non-permafrost regions to evaluate simulated soil carbon stock. While the combined 0–3 m dataset is still not sufficient to evaluate the deep carbon stored in Yedoma (~20 m) and peatlands (up to 10 m), to our knowledge it remains the deepest available gridded SOC product.

For 0-2 m soil layers, the new spin-up generally reproduces the observation-based spatial distribution of total OC. However, it overestimates OC by 16-40 kg C m$^{-2}$ across northern North America and Europe, the Yedoma region, and eastern Asia, while underestimating OC by 8-24 kg C m$^{-2}$ in East Siberia (Fig. 8c). As a result, the simulated total OC for 0-2 m soil layers over the Northern Hemisphere (> 30°N) from the new spin-up is 1,450 Pg C, 79 Pg C higher than the combined observation-based soil datasets (Table 3). Despite the spatial discrepancies, the difference in total OC between our simulation and the combined observation-based datasets is well within the uncertainty range of WISE30sec (± 215 Pg C) and NCSCD (± 200 Pg C).

For 2-3 m soil layers in permafrost regions, the new spin-up underestimates total OC from NCSCD by 8-16 kg C m$^{-2}$ (~80%) across nearly all permafrost regions (Fig. 8f). Although the inclusion of Yedoma carbon and the higher OC for conventional and peat PFTs at high latitudes in the new spin-up (Fig. 7c) reduces the underestimation of OC from the old spin-up (Fig. S7), the total OC stock for 2-3 m soil layers in permafrost regions from the new spin-up is still only about one-third of that in the NCSCD (92 Pg C vs. 292 Pg C). This suggests that significant uncertainties remain in simulating vertical distribution of soil carbon with MICT, or some other processes such as thermokarst deposits (Anthony et al., 2014) or deltaic deposits (Tarnocai et al., 2009) are still needed to be incorporated in the model. However, it is also important to note that the number of high-latitude pedon samples located at sites in northern North America and Siberia and extending down to 3 m are quite limited in WISE and NCSCD (Batjes, 2016; Hugelius et al., 2014), which could also contribute to the discrepancies between the observation-based and simulated soil carbon.

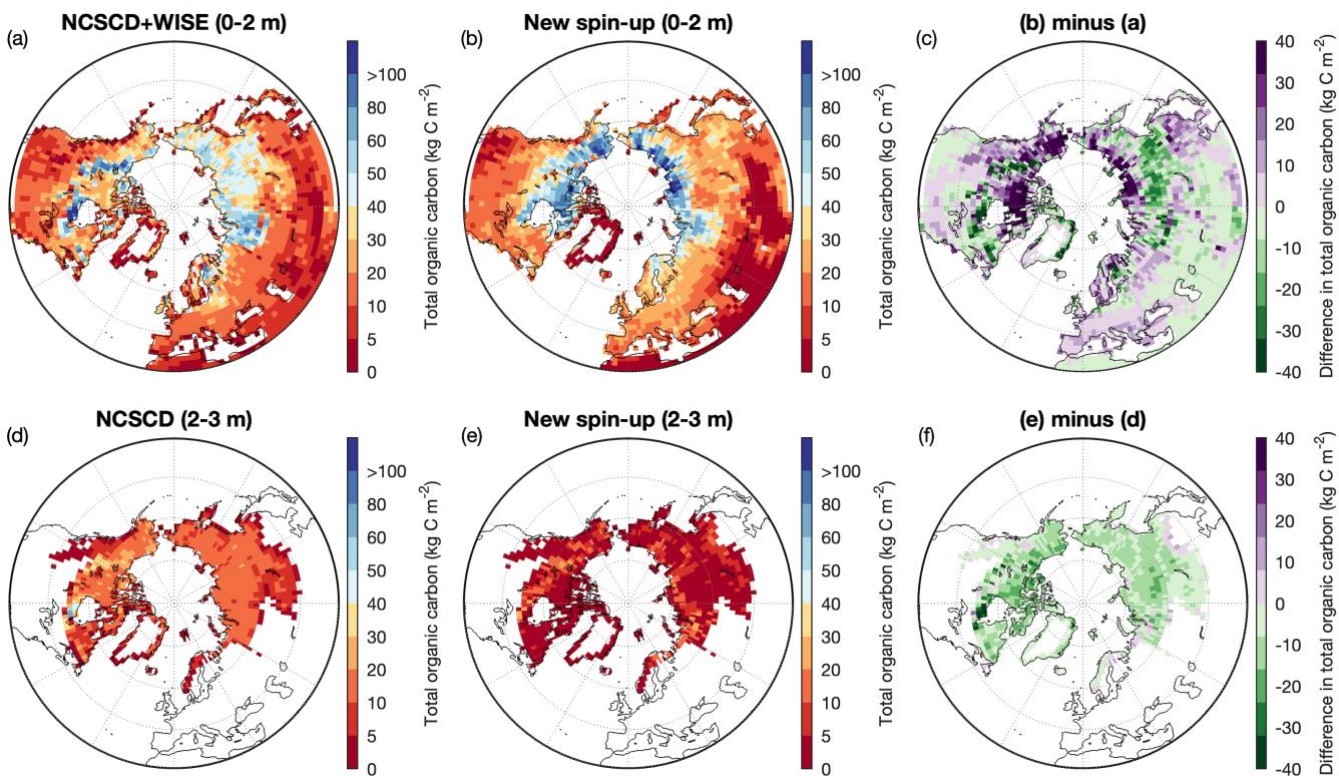

**Figure 8: Comparison of spatial distributions of total organic carbon for 0-2 m and 2-3 m soil layers over the Northern Hemisphere (> 30°N) between two observation-based soil datasets (NCSCD for 0-3 m soil layers in permafrost regions, and WISE for 0-2 m soil layers in non-permafrost regions) and the new spin-up simulation. The first two columns show total organic carbon from the two observation-based soil datasets ((a) and (d)) and new spin-up simulations ((b) and (e)), and the third column for the difference between simulations and observations.**

## 4 Discussion

### 4.1 Interpretation of differences between the old and new spin-up simulations

Compared to the old spin-up simulation, the new spin-up protocol presented here incorporates Yedoma carbon and a 'realistic inception history' of peatland evolution from 16 ka to the present. As a result, the simulated results add 157 Pg C of Yedoma carbon to the total SOC stocks and reproduce the observation-derived spatial patterns and vertical distributions of Yedoma carbon (Fig. 4). For the peat PFT, due to the shorter duration of peat carbon accumulation in the new spin-up, the simulated deep SOC stocks and the passive SOC pool reduce by 20 Pg C (7%) and 35 Pg C (43%), respectively (Table 3), with the reduction being more obvious in younger peatlands (Fig. 6). However, the differences between the two spin-up simulations go beyond this.

In the old spin-up, the SubC is run once for 13,500 years, whereas the new spin-up divides this time span into nine simulation periods. As described in the Methods section, the SubC run uses constant boundary conditions including monthly litter input, soil moisture, and soil thermal conditions from the last year of the preceding FullO simulation. In other words, the 13,500-year SubC run in the old spin-up uses soil temperature ($T_{soil}$) in a low-SOC-stock context of 287 Pg C (Fig. 5c). In contrast, in the new spin-up, the accumulation of OC during each kyr feeds back into $T_{soil}$, leading to a lower $T_{soil}$ in the following kyr due to the insulation effect of SOC, especially in high latitudes (Fig. 7). This lower $T_{soil}$ reduces soil carbon decomposition, promoting carbon accumulation in the new spin-up simulation. The SOC-$T_{soil}$ positive feedback (more soil carbon leads to cooler soils and increased SOC accumulation) is an important driver of higher SOC stocks derived from conventional PFTs in the new spin-up (Table 3 and Fig. S4).

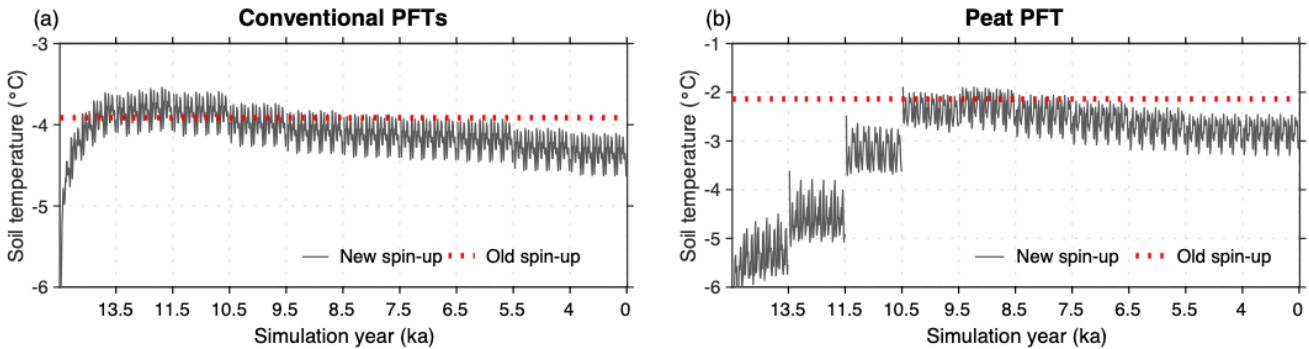

**Figure 9: Comparison of annual soil temperature for 0-20 cm for conventional PFTs (a) and peat PFT (b) in the Arctic region (60-90°N) between the new and old spin-up simulations. We only show the soil temperature used in the SubC run for the old spin-up, i.e., the fixed value from the last year of preceding FullO simulation as background (the red dashed line) for comparison.**

The $T_{soil}$ for 0-20 cm over peatlands north of 60°N in the new spin-up is also lower than in the old spin-up in high latitudes due to the same SOC-$T_{soil}$ positive feedback (Fig. 9b), which partly explains the higher SOC in 0-2 m peat soils and in the Arctic region despite a shorter duration of carbon accumulation in the new spin-up (Table 3 and Fig. S5). Another process contributing to higher SOC stocks and deeper SOC depth (Fig. S5) in the new spin-up is the mechanism of carbon distribution due to grid-specific changes in PFT fraction. As explained in Section 2.2, in MICT, carbon from shrinking PFTs is redistributed to expanding PFTs to ensure carbon conservation. Thus, when the peat PFT of a given grid cell expands in the new spin-up, it inherits SOC from conventional PFTs. Since conventional PFTs in high latitudes have high SOC density and deep SOC depth up to 3 m (Fig. 7 and Fig. S4), the expanding peatlands in the new spin-up could exhibit higher SOC and deeper SOC depths than those in the old spin-up (Fig. S5). While in middle latitudes, where the SOC inherited from conventional PFTs is modest, the peat PFT shows a 53 Pg C (18%) lower SOC, highlighting the important impacts of representing the peatland inception time in the model. Moreover, since all peatlands use the constant observation-derived constraint for the maximum carbon density in each soil layer ($C_{obs,l}$ in Eq. (1)), high-carbon-input peatland areas in the model could accumulate very deep SOC.

For example, in some areas of East Siberia and Alaska, the maximum SOC depth for peat PFT in the old spin-up reaches 38 m (Fig. S5d), implying limitations in applying current parameterization of peat carbon accumulation for all peatlands. By constraining the inception age of peatlands in the new spin-up, the shortened duration of carbon accumulation significantly reduces SOC depth for some young peatlands within these regions (Fig. S5 e-f).

Regardless of whether peatlands are initiated simultaneously or spatially variably, the simulated total northern peatland carbon stock (~630 Pg C) is at the high end compared to previous estimates, e.g., $547 \pm 74$ Pg C from Yu et al. (2010), $415 \pm 150$ Pg C from Hugelius et al. (2020), 410 (315–590) Pg C from Treat et al. (2019), and 463 Pg C from the earlier ORCHIDEE-MICT-PEAT version in Qiu et al. (2019). This overestimation may be also attributed to the SOC-$T_{soil}$ feedback (or coupling). As

evaluated in Xi et al. (2024), wetter and higher-SOC peat soils tend to maintain lower temperatures than the grid-cell average. When SOC-$T_{soil}$ coupling is implemented at the PFT level (i.e., tiled coupling) in the base model (ORCHIDEE-MICT-teb), the simulated peat SOC stock increases to 620 Pg C, compared to 534 Pg C with grid-cell-level coupling, i.e., an 86 Pg C difference. This suggests that further evaluation and calibration of peat SOC in the improved model version are still needed in the future.

## 4.2 Implications and limitations

Incorporating high-latitude deep carbon into LSMs or ESMs is needed to quantify its response to climatic change, given the vast carbon storage and the accelerated warming in these regions. However, it remains inadequately addressed, or even entirely absent, in current ESMs (Schädel et al., 2024). Although far from complete, the improvements to soil carbon accumulation

from this study, compared to the original MICT, offer an acceptable model tool capable of simulating both shallow and deep soil carbon for conventional soils, peatlands, and Yedoma deposits consistently. The next step in application of this model will be to investigate if the inclusion or modification of deep carbon in the new spin-up will impact projected soil carbon emissions and whether this will alter the shift timing of the terrestrial transition from net carbon sink to carbon source in the years to come (Koven et al., 2011, 2015; Nitzbon et al., 2024). This is a critical scientific question in permafrost carbon feedback

research, as the highly confident projections of active layer thickening under future climate change (IPCC, 2021) are expected to increase the exposure of deep carbon to higher temperatures, thereby accelerating permafrost carbon loss. Importantly, none of the ESMs used in the current IPCC AR6 report explicitly simulate deep carbon dynamics (IPCC, 2021).

Additionally, an interesting finding from this study is that the improvements of deep carbon representation alter the distribution

of simulated SOC quality, i.e., the fractions of active, slow, and passive SOC (Fig. 10). In the CENTURY model, decomposed active SOC is partially released as $CO_2$ and partially transferred to slow and passive carbon pools. Similarly, decomposed slow SOC partly moves to the passive carbon pool. Since the additional SOC in the new spin-up is concentrated in very high latitudes (Fig. 7c), where the cold environments inhibit the SOC flow from active and slow pools to the passive pool, this results in 2%

and 4% higher active and slow SOC, respectively but 6% lower passive SOC over the Northern Hemisphere (> 30°N) in the new spin-up (Fig. 10). The higher fractions of active and slow SOC are more evident in higher latitudes. In areas north of 70°N, the new spin-up shows 3% and 7% higher active and slow SOC, respectively, compared to the old spin-up (Fig. 10). As deep carbon becomes increasingly exposed to a warming-induced thickening of the active layer, the higher fractions of active and slow SOC in the high latitudes, as well as in northern peatlands as mentioned before (Fig. 6) simulated by the new spin-up could amplify the projected increase in soil carbon emissions.

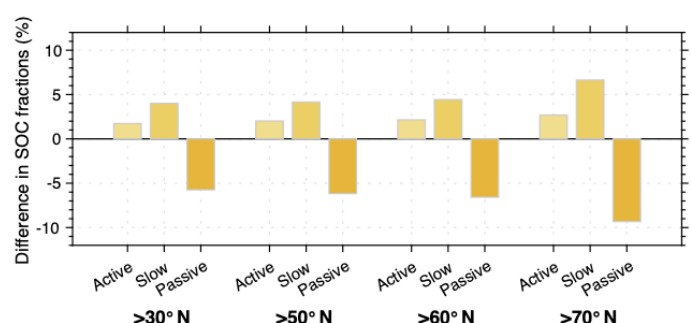

**Figure 10: Difference in SOC fractions between the new and old spin-up simulations for areas > 30°N, > 50°N, > 60°N, and > 70°N. The four areas include 100%, 79%, 55%, and 11% of total SOC of the study area, respectively in the new spin-up simulation.**

This study contains several potential limitations. First, we only simulated the sedimentation process over present-day Yedoma deposits, while the formation and evolution of Yedoma deposits has taken place throughout the last glacial-deglacial transition. Coupling with paleoclimate models and applying the sedimentation-enabled model to specific epochs could help reveal the role of Yedoma deposits in the variation of atmospheric $CO_2$ concentration during the last glacial-deglacial transition (Ciais et al., 2012). Second, in simulating peat carbon accumulation since 16 ka, we used recycled 1901-1920 climate data to force the model in the new spin-up, in order to compare with the old spin-up simulation. Using the actual climate data could increase carbon accumulation during the cold pre-Holocene deglacial period, reduce accumulation during the warm mid-Holocene, and increase accumulation again during the cold late-Holocene (Kaufman et al., 2020; Osman et al., 2021). Coupling with paleoclimate models (Kleinen et al., 2012, 2016; Treat et al., 2019) or using available datasets such as TraCE-21ka (He et al., 2013) would provide a more realistic carbon accumulation history for peatlands. Third, as we used present-day peatland distributions to reconstruct Holocene peatland maps, our simulations do not account for peatlands that were historically drained for human land-use. Within our study region this is particularly relevant for Northern Europe, where substantial peatland drainage for agriculture and forestry took place over the past two centuries (Fluet-Chouinard et al., 2023). A previous study has shown that drainage or peatland-to-cropland conversion during the years 850-2010 could increase total historical carbon stocks in northern peatlands by 72 Pg C (Qiu et al., 2021). However, because the development presented in this study is primarily aimed at projecting future changes in deep carbon stored in peatlands. Omitting historical carbon losses from

drainage is likely to have a relatively limited influence on future projections, as those losses mostly contributed to past emissions.

**5 Conclusion**

This study presents a new, more realistic, spin-up protocol for high latitude carbon formation over the late Pleistocene and
560 Holocene by representing / improving deep carbon accumulation in Yedoma deposits and peatlands in the ORCHIDEE-MICT land surface model. Compared to the existing spin-up approach, the simulation outputs of the new scheme add 157 Pg C to the present-day Yedoma region, while reducing deep carbon below 3 m by 20 Pg C and passive soil carbon by 35 Pg C in northern peatlands due to the shorter duration of peat carbon accumulation after constraining peat initiation time in the model. Since the inclusion of SOC-$T_{soil}$ feedback in carbon accumulation simulation slightly increases the carbon stock in conventional soils
by 68 Pg C, the total organic carbon stock across the Northern Hemisphere (> 30°N) simulated by the new spin-up is 2,028 Pg C, which is 226 Pg C higher than the old spin-up. Although evaluation against an observation-based soil dataset reveals significant challenges in simulating soil carbon for 2-3 m with ORCHIDEE-MICT, the improvements in carbon accumulation modeling from this study provide a model version for prediction of deep soil carbon evolution during the last glacial-deglacial transition, and its response to future warming.

**Code availability**. The ORCHIDEE-MICT-teb-deepc model (r8704) code used in this study is open source and distributed under the CeCILL (CEA CNRS INRIA Logiciel Libre) license. It is deposited at https://forge.ipsl.jussieu.fr/orchidee/wiki/GroupActivities/CodeAvalaibilityPublication/ORCHIDEE-MICT-teb-deepc (last access: 22 October 2024) and archived at https://data.ipsl.fr/catalog/srv/eng/catalog.search#/metadata/704f7a4e-3a1d-4a51-
575 8c70-1920c32684ce and https://zenodo.org/records/15306029 (last access: 4 May 2025) (Xi, 2025a), with guidance to install and run the model at https://forge.ipsl.jussieu.fr/orchidee/wiki/Documentation/UserGuide (last access: 22 October 2024). Due to limitations imposed by their size, the model input files have not been uploaded to the public repository; however, they can be accessed from the corresponding author upon reasonable request. The Yedoma map can be obtained from https://apgc.awi.de/dataset/iryp-v2 (last access: 22 September 2023) (Strauss et al., 2021). The radiocarbon dates of peat cores
can be obtained from https://doi.pangaea.de/10.1594/PANGAEA.902549 (last access: 15 September 2023) (Loisel et al., 2017). The initial WISE30sec dataset can be obtained from https://data.isric.org/geonetwork/srv/api/records/dc7b283a-8f19-45e1-aaed-e9bd515119bc (Batjes, 2016). The initial NCSCD dataset can be obtained from https://bolin.su.se/data/ncscd/ (Hugelius et al., 2013). The resampled data of WISE30sec and NCSCD used in this study and codes to process data, generate all results, and produce all figures are archived at https://zenodo.org/records/15371113 (last access: 9 May 2025) (Xi, 2025b).

**Author contributions.** YX designed the new spin-up protocol and implemented the model development. PC, DZ, CQ, YZ, SP, and GF provided general scientific guidance to improve the research and interpret results. YX performed the simulations, did the analysis, and wrote the paper. All authors contributed to commenting on and writing the manuscript.

**Competing interests.** The contact author has declared that none of the authors has any competing interests.

**Acknowledgements.** The authors are grateful to the ORCHIDEE group for their kind help on coding, preparing input files, and publishing the model version on svn.

**Financial support.** This research was supported by the CALIPSO (Carbon Loss in Plant Soils and Oceans) project, funded through the generosity of Eric and Wendy Schmidt by recommendation of the Schmidt Futures program.

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
