# Peer review of "Representing high-latitude deep carbon in the pre-industrial state of the ORCHIDEE-MICT land surface model (r8704)"

_Geoscientific Model Development, 2024_

## Author Response (AR1)

**Response to the editor**

**General comments**

*Dear authors,*

*Unfortunately, after checking your manuscript, it has come to our attention that it does not comply with our "Code and Data Policy".*
*https://www.geoscientific-model-development.net/policies/code_and_data_policy.html*

*You have archived the ORCHIDEE code in a site that does not comply with our requirements. Therefore, you must store the code in a repository that complies with our policy, and provide a link and permanent identifier (e.g. DOI or Handle).*

*A similar issue happens with the WISE30sec and the NCSDC data. It would be good if you could contact their authors/maintainers and request them to store the data in an acceptable repository, and then reply to this comment with the information.*

*Please, reply to this comment as soon as possible with the information about it, and modify the Code Availability section of your manuscript accordingly in future versions of your manuscript.*

*Juan A. Añel*

**[Response]** Thank you for pointing out this important issue. We have archived the model code in a public repository on Zenodo (https://zenodo.org/records/15306029) and cited it in the revised manuscript to ensure reproducibility.

Regarding the WISE30sec and NCSCD datasets, we did not request the original authors to deposit them in an alternative repository, as these datasets have been widely used over the past decade and are consistently cited using the established links. Changing those links might disrupt continuity for previous studies. Instead, to support transparency and reproducibility, we have uploaded the subset of these datasets used in our study (limited to our study domain and spatial resolution), along with all processing scripts and code used to generate the results and figures in our study, to a separate public Zenodo repository (https://zenodo.org/records/15371113). This has also been cited in the revised manuscript.

If this approach does not meet the data availability standards, we will try to reach out to the original dataset authors for any possibility of archiving the data in a compliant repository, though we cannot guarantee a positive answer.

We hope this solution is acceptable, and we appreciate your understanding.

**Response to the reviewer 1**

**General comments**

*This paper introduces new model processes to represent old and deep carbon formation in Yedoma and peatland soils. It is an interesting model development, as it is a common issue for many land surface or ecosystem models to simulate these deep and old carbon dynamics explicitly. However, I struggled to understand the model comparisons, evaluation, and calibration due to the current structures and the scattered information.*

**[Response]** Thank you very much for your time and effort in reviewing our manuscript, and for your constructive feedback on our work. We greatly appreciate your thoughtful comments, which have helped us improve both the clarity and structure of the paper. Following your suggestions, we have made the following revisions:

(1) Added summary paragraphs describing the model merging strategy and revised Fig. 1 to provide a clearer graphical overview of the approach, helping readers better understand the overall model development and the rationale for focusing on the evaluation of simulated carbon accumulation.

(2) Reorganized the Results and Discussion sections, and added the explanation of our evaluation strategy to improve the presentation of model comparisons, evaluation, and validation.

(3) Provided additional details on the peatland and Yedoma PFTs, and included a new subsection titled *"Changes in plant and soil carbon pools following conversion from conventional soils to peatlands"* in the Data and Methods section, to clarify the mechanisms underlying changes in peatland areas and peatland inception timing.

Please find our point-by-point responses to each of your comments below.

**Major comments**

1. *The authors merged different versions of ORCHIDEE models, so it seems unclear if any independent calibration or evaluation has been conducted before comparing carbon accumulations to ensure the model's performance is on the right track before adding new processes.*

**[Response]** Thank you for raising this important point. In this study, we merged two model versions: ORCHIDEE-MICT-teb (Xi et al., 2024) and ORCHIDEE-MICT-Yedoma (Zhu et al., 2016). The core structure of the improved model is based on ORCHIDEE-MICT-teb, a revised version of the widely used ORCHIDEE-MICT branch (e.g., contributions to the Global Carbon Budget (Friedlingstein et al., 2022), ISIMIP3a simulations, etc.) which targets high-latitude processes (Guimberteau et al., 2018). Compared to the original ORCHIDEE-MICT version, ORCHIDEE-MICT-teb incorporates an improved representation of tiled energy budgets, and has been comprehensively evaluated in our previous work (Xi et al., 2024), including its performance in simulating energy (their Figs. 13–17),

carbon (their Fig. 12), and hydrological processes (their Fig. 15) at monthly timescales and across the North hemisphere.

The Yedoma-specific additions from ORCHIDEE-MICT-Yedoma were integrated to simulate carbon accumulation processes specific to Yedoma regions. Because these additions primarily influence soil carbon dynamics, our evaluation in this study (Section 3.1) focuses on carbon-related outputs in Yedoma regions. Given the relatively limited spatial extent of Yedoma (0.47 Mkm$^2$, 0.7% of areas north of 30°N), our model modifications are expected to have only a minor effect on other regions, and the results from the precursor version hold in this study.

Regarding the peatland processes, the base version already includes peatland carbon accumulation. We did not introduce new process developments, but we significantly revised the way peatland initiation is simulated. Instead of assuming simultaneous peatland formation across all northern peatlands, our new approach uses observation-based, spatially varying, peatland age maps to prescribe the onset of peat formation at each grid cell. This was achieved by prescribing peatland cover maps at different epochs during the Holocene, which the model reads to simulate peatland expansion over time. This approach resolves effects of observed variations in initiation timing on long-term peat carbon accumulation. Since the dataset we used to prescribe peat initiation includes all / most known observations, peat initiation cannot be separately evaluated. We did not further evaluate the hydrological and energy processes of peatlands, as the total simulated peatland soil carbon remains very close to that of the base version (631.4 Pg C vs. 632.0 Pg C), suggesting only minor difference between model versions on regional scale. The primary change of introducing spatially varying peatland initiation lies in the partitioning between fast- and slow-turnover soil carbon pools (see our response to your detailed comment #1 for more details on the soil carbon module), which we have evaluated in detail in Section 3.2.

In response to your comments, we have added two paragraphs at the beginning of Section 2 *Data and Methods* (L101-123, copied as below) to summarize our model development strategy, including the details of each model version and our previous evaluation work. In addition, we have revised Fig. 1 (shown as Fig. R1 below) to provide a clearer graphical summary of our approach, and have cited Fig. 1 in the text where the old and new spin-ups are introduced. We believe these changes will help readers, particularly those less familiar with model development, better understand which aspects of the improved model require evaluation.

"**To implement the model development, we used a branch version of ORCHIDEE-MICT, namely ORCHIDEE-MICT-teb (r8205), from Xi et al. (2024) as our base model. This version has PFT-specific (plant functional type-specific) energy budgets, soil thermics, and their interactions with carbon and hydrological processes, and has been comprehensively evaluated for its performance in simulating carbon, energy, and hydrological processes (Xi et al., 2024). To simulate Yedoma**

carbon accumulation, we incorporated sedimentation processes from Zhu et al. (2016), adapting them to be compatible with the PFT-specific framework of the new model. The model version used by Zhu et al. (2016) relied on grid-cell-averaged energy budgets, which meant that Yedoma carbon was mixed with coexisting PFTs (bare soil, tree, grass, crop) to regulate the average energy budget. Given that Yedoma soils typically have much higher carbon densities than general soils, accounting for their distinct carbon-energy-water feedbacks such as the thermal insulation effect of SOC (Koven et al., 2009; Zhu et al., 2019) is essential for realistically simulating both their historical development and future evolution. Regarding peatland processes, the base version already included peatland carbon accumulation. We thus did not develop new peatland processes, but we revised the way peatland initiation is simulated during the Holocene. Instead of assuming a homogeneous peatland age across northern peatlands starting in early Holocene, we employed spatially explicit peatland age maps based on observational data to define the onset of peat development at each grid cell. This was achieved by prescribing peatland cover maps at different epochs during the Holocene, which the model uses as a forcing to simulate realistic peatland expansion over time. As a result, the improved model allows for the independent simulation of Yedoma carbon, peatland carbon, and conventional soil carbon dynamics within a single grid cell.

In Section 2.1, we describe the soil carbon models in detail, including the general soil carbon processes and the peatland carbon accumulation scheme in the base version (Xi et al., 2024), as well as the Yedoma carbon accumulation processes from Zhu et al. (2016). Section 2.2 then presents the setup of the carbon accumulation simulations, using both the base version (without Yedoma carbon accumulation and with uniform peatland inception timing) and the improved version (with Yedoma carbon accumulation and spatially varying peatland inception)."

[Figure]

Fig. R1 (also shown as Figure 1). Schematic representation of soil carbon accumulation across different simulation stages in the old and new spin-up simulations. The old spin-up uses the ORCHIDEE-MICT-teb (MICT-teb) model version, includes 16 plant functional types: 15 conventional PFTs (bare soil, 8 tree PFTs, 4 grass PFTs, and 2 crop PFTs) and one peatland PFT (C3 grass) within a grid cell (Table S1). In the old spin-up, the cover fraction of peatland is fixed throughout the simulation, and all peatlands are assumed to have formed simultaneously, 13,500 years ago. By contrast, the new spin-up merges ORCHIDEE-MICT-Yedoma (MICT-Yedoma) with MICT-teb, enabling the model to simulate Yedoma carbon accumulation during the late Pleistocene and subsequent carbon accumulation in peatland and conventional soils during the Holocene. The new spin-up includes the same 15 conventional PFTs, one peatland PFT, and an additional Yedoma PFT (C3 grass) within a grid cell (Table S1). Peatland cover varies dynamically across nine Holocene epochs (shown in the peat PFT domain), based on prescribed maps constructed from peat basal age observations and present-day peatland distribution.

2. *It is a bit confusing for me* *what has been developed in this study in terms of yedoma deposits.* *In the method section, it states that this study merged yedoma process from Zhu et al., 2016, and then in the Results section on L275, the authors used the same survey to evaluate the model performance. In Figure 4,* *why not compare the simulation outputs between the old and new spinup to show the model differences?*

**[Response]** As mentioned in our last response to your comment, for Yedoma deposits, we merged processes from Zhu et al. (2016), but adapted them to be compatible with the PFT-specific framework

in the base version (ORCHIDEE-MICT-teb). We did not directly compare simulation outputs between the original and improved versions for two main reasons:

(1) Model structure changes: While the conceptual representation of Yedoma-specific processes remains similar, their implementation differs due to structural updates in the improved version. In Zhu et al. (2016), energy budgets were calculated at the grid-cell level, whereas the improved version employs PFT-specific energy budgets. This shift affects not only carbon accumulation but also soil thermal and hydrological processes, particularly in high-carbon-density Yedoma soils. Furthermore, ORCHIDEE-MICT-teb includes several updates and bug fixes not present in the Zhu's version. For example, we used a newly introduced PFT (boreal C3 grass) to better represent Yedoma-region vegetation, while Zhu et al. relied on a generic C3 grass type. This change alters vegetation parameters and may influence simulated carbon dynamics. Accurately isolating the specific impact of our developments would require substantial effort to ensure strict comparability between versions, which is not justified given the high workload.

(2) Lack of original outputs: The simulation output files from Zhu et al. (2016) are no longer available, which prevents direct comparison of simulation outputs between the original and improved versions. Although we still have access to the model code, reconstructing all the input files and rerunning the outdated version would be time-consuming and of limited value, especially given that an improved model version is already available.

As a result, instead of a direct comparison, we reproduced similar figures to those in Zhu et al. (2016) and used the same survey to evaluate the model performance, an indirect way to compare the two versions. Despite structural and parameterization differences, our simulation produced similar results: a total soil carbon stock of 141 Pg C and an average Yedoma soil carbon depth of ~20 m, closely matching the 125-145 Pg C and ~20 m depth reported by Zhu et al. (2016). This consistency suggests that the core mechanism driving vertical carbon accumulation (Eq. (2)), governed by measurement-derived deposition rates, remains robust and effective in capturing Yedoma soil carbon dynamics. Also, it indicates that our improved model version successfully integrates Yedoma-specific processes without conflating them with existing components of the base model (ORCHIDEE-MICT-teb).

We did not add further explanations on this point in the revised manuscript, given that including overly detailed descriptions of the model merging might distract from the overall understanding of the model developments. However, together with your next comment, we have reorganized the Results and Discussion sections and added a paragraph at the beginning of Section 3 (L317-323, copied as below) to summarize and clarify our evaluation strategy.

**"In this section, we present the simulated Yedoma carbon accumulation from the improved model version in Section 3.1. In Section 3.2, we examine how peatland soil carbon accumulation responds to the implementation of spatially varying peatland initiation timing. Since the vertical carbon**

**transfer process in Yedoma is driven by deposition rates derived from site-level measurements, and peatland development is informed by prescribed maps extrapolated from point-based peat age data, we evaluate model performance by comparing simulated soil carbon from both the old and new spin-up schemes, as well as against site-level observations. In Section 3.3, we provide a spatial evaluation using the deepest available gridded soil carbon map (to our knowledge), although it only covers the top 0-3 m of soil.**"

> 3. *When comparing the spatial distribution of simulated carbon accumulation and maximum depth, why not compare the existing datasets to illustrate whether the changes make the simulations closer to the observation-based patterns? I later saw the comparison of the new spinup output with the existing database in Figure 9 in the Discussion section, which makes it difficult to judge the performance of these new model developments.*

**[Response]** Thank you for your comment. In our previous Results section, we showed only the spatial distribution of simulated Yedoma carbon accumulation and maximum depth, without comparing with existing observation-based spatial patterns. Our logic was that, since we used deposition rates derived from site-level measurements (1.0 mm yr$^{-1}$) and a specified accumulation period (19,000 years), the most direct validation was to test whether the model—driven by dynamic carbon cycle processes, observation-based deposition rates, and the specified duration—could reproduce the observed Yedoma carbon depths and total carbon stocks.

In Fig. 9, we used a gridded SOC map that combines WISE30sec (global soils, 0-2 m) (Batjes, 2016) and NCSCD (northern permafrost regions, 0-3 m) (Hugelius et al., 2013) datasets to evaluate our model's performance in simulating soil carbon within the top 3 m only. To our knowledge, this is currently the deepest available gridded SOC map. However, it remains insufficient for representing deep carbon in Yedoma and peatlands which reach depths of up to ~20 m and ~10 m, respectively. Our model improvements were explicitly designed to capture such deep carbon storage. Therefore, using a 0-3 m SOC dataset for evaluation is not sufficient enough. In the absence of gridded observational data for deeper SOC deposits we argue that the dedicated site-level deep soil core measurements which we used offer more appropriate and informative benchmarks for assessing the performance of our improved version.

In the revised manuscript, we still provide a spatial comparison using the 0-3 m SOC map at the end, along with a note on its limitations in representing deep soil carbon stocks. In response to your last comment, we have added a paragraph at the beginning of Section 3 (L317-323) to summarize and clarify our evaluation strategy. In addition, we revised the title of Section 3 as "**Results and evaluation**", and moved the evaluation against the spatial SOC map from the previous Discussion section into this Section, as Section 3.3 (L424-457). We apologize again that our previous manuscript did not clearly

separate the Results and Discussion sections, which may have made our evaluation strategy difficult to follow.

4. *The authors may want to consider* merging the Results and Discussion sections, *as some content in the Discussion were in the Results section and vice versa.*

**[Response]** We agree and following our response to your major comments #2 and #3, we have merged the simulated results and their evaluation in Section 3 of the revised manuscript. Section 4 is now dedicated to the discussion, including Section 4.1, which provides a further interpretation of the differences between the old and new spin-up simulations, and Section 4.2, which discusses the implications and limitations of our development.

5. *What happens in the model (regarding plant and soil carbon pools) when converting an upland cell fraction to a peatland fraction? It could be good to describe so readers can better understand the mechanism of these new changes in dynamic peatland areas and peatland inception time.*

**[Response]** Thank you very much for this constructive suggestion. In the ORCHIDEE-MICT model, when an upland cell fraction is converted to a peatland cell, the new peatland fraction first inherits the plant and soil carbon pools from the displaced upland cell in order to conserve mass. After this transition, the newly formed peatland begins to grow peatland-specific PFT and accumulate soil carbon according to peatland soil characteristics. Compared to conventional PFTs (e.g., forest or grass), the peatland PFT features a shallower rooting depth and is prescribed with distinct soil hydrological properties, including specific values for hydraulic conductivity and diffusivity. Moreover, the peat soil tile does not allow drainage at the base of soils and receives lateral surface water input from other upland PFTs within the same grid cell. These water-logged soil conditions substantially suppress decomposition, promoting the accumulation of soil organic carbon.

Regarding vertical soil carbon transfer, conventional PFTs accumulate soil carbon to a maximum depth of 3 m through three main processes: (1) root-density-dependent organic carbon inputs, (2) depth-dependent decomposition regulated by vertically stratified soil temperature and moisture, and (3) carbon diffusion via bioturbation by animal and plant activity and via cryoturbation in permafrost soils. By contrast, the peatland PFT employs a more efficient scheme for vertical carbon transfer (as described in the second paragraph of Section 2.1), enabling substantial carbon accumulation down to depths of up to 10 m.

In the revised manuscript, we have added a new subsection in Section 2.2.2 (L303-315, copied as below) to describe the changes in plant and soil carbon pools that occur when a fraction of an upland cell (i.e., conventional soils) is converted into a peatland fraction.

"**• Changes in plant and soil carbon pools following conversion from conventional soils to peatlands**

**In the starting MICT version from Xi et al., (2024), when conventional soils are converted to peatlands within a grid cell, the newly established peatland fraction initially inherits the plant and soil carbon pools from the displaced conventional soils to ensure mass conservation. After this transition, the peatland fraction begins to grow peatland-specific PFT and accumulate soil carbon according to peatland soil characteristics. As described in Section 2.1, compared to conventional PFTs (e.g., forest or grass), the peatland PFT features a shallower rooting depth and is assigned distinct hydrological properties. Moreover, the peat soil tile does not allow drainage at the base of the soils and also receives lateral surface water input from other non-peatland PFTs within the same grid cell. These water-logged soil conditions substantially suppress decomposition, promoting the accumulation of soil organic carbon. Peatland tiles also differ regarding vertical soil carbon transfer. Conventional PFTs can accumulate carbon down to a depth of 3 m through root-distributed inputs, depth-dependent decomposition, and vertical diffusion via bioturbation and cryoturbation (see the first paragraph of Section 2.1). In contrast, the peatland PFT employs a more efficient scheme for vertical carbon transfer (detailed in the second paragraph of Section 2.1), enabling substantial carbon accumulation down to depths of up to 10 m.**"

**Some detailed comments:**

1. *Abstract:*

   *L21: "A passive soil carbon pool" could be rephrased. For Non-specialists in this field, we won't know* what "passive" means here. *It can be easily misunderstood as it is not biologically active or is not contributing to any fluxes. The same goes to L27..What does "less passive" really mean?*

**[Response]** Sorry for this very technical phrasing. In the ORCHIDEE-MICT model, there are three conceptual soil carbon pools, 'active', 'slow', and 'passive' pools, defined by their turnover rates. The turnover time at 5 °C without moisture limitation for the three carbon pools is 0.84, 31, and 1,363 years, respectively. The 'passive' soil pool turns over much more slowly than the 'active' and 'slow' pools, but it is still biologically active and contributes to $CO_2$ fluxes. To improve clarity, we have revised "**a passive soil carbon pool**" as "**a passive soil carbon pool (a conceptual soil carbon pool with longest turnover time)**" at L22 and have revised "**less passive**" as "**a smaller passive soil carbon pool (by 35 Pg C, 43%)**" at L28. Moreover, we have added the specific turnover times of five carbon pools (three soil carbon pools and two litter carbon pools) in the Methods (L127-128), where the soil carbon model is described: "**The turnover times at 5 °C without moisture limitation for the five carbon pools is 0.37, 1.4, 0.84, 31, and 1,363 years, respectively.**"

2. *Figure 1, &L148, what are peat PFT and Yedoma PFT? What are the main differences between these two PFTs? So why only one PFT is allowed to grow on peatland or Yedoma? How PFT chose could influence carbon accumulation over this study's long temporal scales?*

**[Response]** Yedoma is ice-rich, organic-rich permafrost deposits formed during the late Pleistocene, primarily distributed across Arctic Siberia and Alaska (Fig. 1). Peatlands are wetland ecosystems where water-saturated conditions inhibit decomposition, resulting in the accumulation of organic-rich peat layers over thousands of years, mainly distributed in northern high latitudes (Fig. S2p). In our simulations, both the peat PFT and the Yedoma PFT are based on the existing C3 grass PFT (Table S1), with rooting depth as the only structural difference—shallower for the peat C3 grass. This framework allows for further refinement of these PFTs in future developments. The main differences between the two PFTs in current model lie in their associated soil hydrology characteristics and the schemes used for vertical transfer of soil organic carbon, as mentioned earlier and described in Section 2.1. Specifically, peatland soils incorporate specific processes to retain water and maintain water-logged conditions, which are important for peat formation. The vertical transfer of soil carbon in peat and Yedoma soils is simulated by distinct schemes, with Eq. (1) for peat carbon and Eq. (2) for Yedoma carbon, respectively.

For the number of PFTs, it's flexible and technically possible to represent multiple PFTs for peatlands and Yedoma within one grid cell in our simulation. However, the peatland / Yedoma map we used provides only the fractional coverage of peatlands and Yedoma in each grid cell, without specifying vegetation types or biomes. Given that C3 grass is the dominant vegetation type in boreal peatlands and Yedoma regions, we chose to represent each with a single C3 grass PFT in our simulations. We acknowledge that using different PFTs could affect carbon accumulation by influencing several processes, including $CO_2$ assimilation rates, surface characteristics (e.g., albedo and roughness), surface energy budgets, and the associated soil thermal and hydrological dynamics.

To clarify this point, we have added the explanation of each PFT in the caption of Fig. 1 (L187-193). "... **The old spin-up uses the ORCHIDEE-MICT-teb (MICT-teb) model version, includes 16 plant functional types: 15 conventional PFTs (bare soil, 8 tree PFTs, 4 grass PFTs, and 2 crop PFTs) and one peatland PFT (C3 grass) within a grid cell (Table S1). In the old spin-up, the cover fraction of peatland is fixed throughout the simulation, and all peatlands are assumed to have formed simultaneously, 13,500 years ago. By contrast, the new spin-up merges ORCHIDEE-MICT-Yedoma (MICT-Yedoma) with MICT-teb, enabling the model to simulate Yedoma carbon accumulation during the Late Pleistocene and subsequent carbon accumulation in peatland and conventional soils during the Holocene. The new spin-up includes the same 15 conventional PFTs, one peatland PFT, and an additional Yedoma PFT (C3 grass) within a grid cell (Table S1).** ..."

3. *L276, what is the value of total SOC stock from the old spin-up then?*

**[Response]** There's no Yedoma PFT or Yedoma carbon accumulation in the old spin-up, so the value is zero (see Table 3).

4. *L306-307, what are the main reasons that two spinup end with different NEP values?*

**[Response]** The different NEP values between the two spin-up simulations likely result from two main factors:

(1) Differences in PFT composition within a grid cell: The old spin-up includes 15 conventional PFTs and one peat PFT, whereas the new spin-up includes 15 conventional PFTs, one peat PFT, and an additional Yedoma PFT. This leads to different net primary production (NPP) due to variation in vegetation distribution and productivity.

(2) Differences in soil carbon stocks: The new spin-up introduces changes in carbon accumulation, particularly in deep soils, which affects total soil respiration.

The altered balance between NPP and heterotrophic respiration ultimately leads to the difference in NEP.


Our next goal is to use this improved version to assess how deep carbon stored in Yedoma and peatlands responds to future climate warming. It is a critical scientific question in permafrost carbon feedback research, as the highly confident projections of active layer thickening under future climate change (IPCC, 2021) are expected to increase the exposure of deep carbon to higher temperatures, thereby accelerating permafrost carbon loss. Importantly, none of the Earth System Models (ESMs) used in the current IPCC AR6 report explicitly simulate deep carbon dynamics (IPCC, 2021). Therefore, we believe our model development goes beyond a technical merger and constitutes a meaningful scientific advancement. It is both timely and impactful, rather than incremental or outdated.

Together with the other reviewer's comment, we have added two paragraphs at the beginning of Section 2 *Data and Methods* (L101-123, copied as below) to highlight the technical complexity and scientific necessity of our development. In addition, we have revised Fig. 1 (shown as Fig. R1 below) to provide a clearer graphical overview of our model development, where the consistent simulation of soil carbon accumulation across Yedoma, peatlands, and conventional PFTs is shown clearly.

"To implement the model development, we used a branch version of ORCHIDEE-MICT, namely ORCHIDEE-MICT-teb (r8205), from Xi et al. (2024) as our base model. This version has PFT-specific (plant functional type-specific) energy budgets, soil thermics, and their interactions with carbon and hydrological processes, and has been comprehensively evaluated for its performance in simulating carbon, energy, and hydrological processes (Xi et al., 2024). To simulate Yedoma carbon accumulation, we incorporated sedimentation processes from Zhu et al. (2016), adapting them to be compatible with the PFT-specific framework of the new model. The model version used by Zhu et al. (2016) relied on grid-cell-averaged energy budgets, which meant that Yedoma carbon was mixed with coexisting PFTs (bare soil, tree, grass, crop) to regulate the average energy budget. Given that Yedoma soils typically have much higher carbon densities than general soils, accounting for their distinct carbon-energy-water feedbacks such as the thermal insulation effect of SOC (Koven et al., 2009; Zhu et al., 2019) is essential for realistically simulating both their historical development and future evolution. Regarding peatland processes, the base version already included peatland carbon accumulation. We thus did not develop new peatland processes, but we revised the way peatland initiation is simulated during the Holocene. Instead of assuming a homogeneous peatland age across northern peatlands starting in early Holocene, we employed spatially explicit peatland age maps based on observational data to define the onset of peat development at each grid cell. This was achieved by prescribing peatland cover maps at different epochs during the Holocene, which the model uses as a forcing to simulate realistic peatland expansion over time. As a result, the improved model allows for the independent simulation of Yedoma carbon, peatland carbon, and conventional soil carbon dynamics within a single grid cell.

In Section 2.1, we describe the soil carbon models in detail, including the general soil carbon processes and the peatland carbon accumulation scheme in the base version (Xi et al., 2024), as well as the Yedoma carbon accumulation processes from Zhu et al. (2016). Section 2.2 then presents the setup of the carbon accumulation simulations, using both the base version (without Yedoma carbon accumulation and with uniform peatland inception timing) and the improved version (with Yedoma carbon accumulation and spatially varying peatland inception)."

[Figure]

Fig. R1 (also shown as Figure 1). Schematic representation of soil carbon accumulation across different simulation stages in the old and new spin-up simulations. The old spin-up uses the ORCHIDEE-MICT-teb (MICT-teb) model version, includes 16 plant functional types: 15 conventional PFTs (bare soil, 8 tree PFTs, 4 grass PFTs, and 2 crop PFTs) and one peatland PFT (C3 grass) within a grid cell (Table S1). In the old spin-up, the cover fraction of peatland is fixed throughout the simulation, and all peatlands are assumed to have formed simultaneously, 13,500 years ago. By contrast, the new spin-up merges ORCHIDEE-MICT-Yedoma (MICT-Yedoma) with MICT-teb, enabling the model to simulate Yedoma carbon accumulation during the late Pleistocene and subsequent carbon accumulation in peatland and conventional soils during the Holocene. The new spin-up includes the same 15 conventional PFTs, one peatland PFT, and an additional Yedoma PFT (C3 grass) within a grid cell (Table S1). Peatland cover varies dynamically across nine Holocene epochs (shown in the peat PFT domain), based on prescribed maps constructed from peat basal age observations and present-day peatland distribution.

Moreover, we have added more text in Section 4.2 for the implications of our model development (L516-522). "...**The next step in application of this model will be to investigate if the inclusion or modification of deep carbon in the new spin-up will impact projected soil carbon emissions and whether this will alter the shift timing of the terrestrial transition from net carbon sink to carbon source in the years to come (Koven et al., 2011, 2015; Nitzbon et al., 2024). This is a critical scientific question in permafrost carbon feedback research, as the highly confident projections of active layer thickening under future climate change (IPCC, 2021) are expected to increase the**

**exposure of deep carbon to higher temperatures, thereby accelerating permafrost carbon loss. Importantly, none of the ESMs used in the current IPCC AR6 report explicitly simulate deep carbon dynamics (IPCC, 2021).**"

2. *There is a spatial mis-match between the two domains (Yedoma and peatlands) which is confusing and never explicitly stated until Table 3. The peatland model simulation includes not only northern temperate, boreal, and Arctic peatlands but also southern temperate and subtropical peatlands and the simulation extends to 30 N. The Hugelius et al. (2020) peatland extent dataset does not go all the way to 30 N. These southern peatlands are also simulated in response to de-glaciation, but this is not a driving factor in peatland initiation for these peatlands at the southern margin of the temperate zone (Treat et al., 2019). A map of grid-cell level peatland initiation would help to clarify this approach. With the extension of modeling to more southern regions, this starts to get challenging because of anthropogenic peatland drainage during the last millennium (e.g. Fluet-Chouinard et al., 2022) which points out the vast extent of wetlands and peatlands lost to drainage. However, this is not accounted for in the past wetland C extent when it is based on present-day areas from Hugelius. This is very problematic.*

**[Response]** Thank you for these suggested improvements. Regarding the domains of Yedoma and peatlands, they are not spatially matched. Our study domain spans 30°N-90°N. Within this domain, Yedoma, as defined by Strauss et al. (2021), covers a relatively smaller region of 60°N-90°N (Fig. 2), while peatlands, as defined by Hugelius et al. (2020), cover 30°N-90°N (Fig. S2). Because the new spin-up simulation includes two stages, and the carbon accumulation for Yedoma and peatland is simulated independently in the new spin-up, spatial match between the Yedoma and peatland domains is not required. As shown in Fig. R1, in simulation stage 1, we simulate Yedoma carbon accumulation over 60°N-90°N. Then in simulation stage 2, we simulate carbon accumulation for all soil types including peatlands, conventional soils, and Yedoma across 30°N-90°N. In this stage, Yedoma sedimentation processes are turned off, and the grid cells identified as Yedoma retain the carbon accumulated during stage 1. Therefore, between the two simulation stages, it is only necessary to ensure a one-to-one correspondence of grid cells where Yedoma is present; full spatial match between Yedoma and peatland domains is not required. Further details on the two simulation stages are provided in our response to your minor comment #4.

In the revised manuscript, we have clarified the spatial domain for all simulation stages of the old and new spin-ups. Specifically, in the introduction of the old spin up, we have included "**All the simulations were run for the Northern Hemisphere (30°N-90°N) at a 2° × 2° spatial resolution.**" (L208). For Yedoma simulation in the new spin-up, we have included "**Consistent with the old spin-up simulation, the Yedoma simulation all new spin-up simulations was also run at a 2° × 2° grid**

**resolution, but only for 60°N-90°N to save computation cost, given the relatively smaller spatial extent of Yedoma deposits (Fig. 1; Strauss et al., 2021).**" (L237-239). In the caption of Fig. 2 (showing the spatial pattern of Yedoma deposits from Strauss et al. (2021)), we have included "**The spatial extent of Yedoma in the map covers only areas north of 60°N.**" (L248). In the captions of Fig. S1 (spatial pattern of peatland basal ages from Loisel et al., (2017)) and Fig. S2 (spatial pattern of reconstructed peat area maps during nine Holocene epochs), we have noted the spatial extent is 30°N-90°N. Finally, in the introduction of peatland simulation in the new spin-up, we have added "**All the simulations for this phase were run for 30°N-90°N at a 2° × 2° spatial resolution, consistent with the old spin-up simulation.**"(L286-287).

Regarding southern peatlands, the Hugelius et al. (2020) peatland dataset covers the area north of 23°N (see Fig. S2 and their Fig. 1A); therefore, our simulations do include 'southern' peatlands all the way down to 30°N. Regarding your request for maps of grid-cell level peatland initiation, these are already shown in Fig. S2 of our original manuscript.

Regarding anthropogenic drainage with the extension of modeling to more southern regions, this is indeed an important point. We agree with the reviewer that the past wetland C extent could not be accounted for when we used present-day peatland areas to reconstruct the Holocene peatland maps. According to Fluet-Chouinard et al. (2023), 0.51 million km² (11%) of global peatlands have been lost to drainage since 1700, with Northern Europe and Southeast Asia (Indonesia and Malaysia) being the regions most affected. Therefore, not accounting for drainage likely leads to an underestimation of historical peatland carbon extent, particularly in Northern Europe, in our study. However, it is important to note that  Fluet-Chouinard et al. does not provide mapped data of reconstructed peatland drainage. Instead, it relies on an external peatland dataset (Xu et al., 2018) and allocates total wetland loss proportionally to peatlands (Fluet-Chouinard et al., 2023). If a detailed and reliable peatland drainage map was available, we could integrate it into the reconstruction of dynamic Holocene peatland maps and re-run our simulations. Given current data limitations and uncertainties, we have chosen to acknowledge the potential underestimation of historical peatland carbon stocks in Northern Europe in the Discussion section of our study.

Qiu et al. (2021) used an earlier ORCHIDEE-MICT version to estimate the change in total historical peatland carbon stocks when accounting for drainage / peatland-to-cropland conversion, and found that northern peatlands converted to croplands emitted 72 Pg C over the period 850-2010 (Qiu et al., 2021). Their simulations activated the dynamic peatland module and assumed three potential scenarios of peatland-to-cropland conversion based on annual cropland maps. However, this approach was not feasible in our study, as we prescribed peatland fractions using reconstructed Holocene peatland maps. In a single simulation, the dynamic peatland modeling and prescribed land cover cannot be used

concurrently. While coordinating peat age information with cropland expansion could improve estimates of historical changes in peatland carbon stocks, it is beyond the scope of our current study, which is primarily aimed at projecting future changes in deep carbon stored in peatlands. Since historical carbon losses from peatland drainage mostly contributed to past carbon emissions, we expect their omission to have a limited impact on future projections.

As a result, we have chosen to keep our current simulation results, but have added further discussion in the revised manuscript (L549-557, copied as below) to acknowledge that our simulation could underestimate historical peatland carbon stocks due to the exclusion of peatland that have been drained. We also note that, if a reliable peatland drainage dataset becomes available in the future, it could be incorporated into our prescribed land cover maps, if necessary for specific research objectives, to better account for this missing past carbon component.

"**Third, as we used present-day peatland distributions to reconstruct Holocene peatland maps, our simulations do not account for peatlands that were historically drained for human land-use. Within our study region this is particularly relevant for Northern Europe, where substantial peatland drainage for agriculture and forestry took place over the past two centuries (Fluet-Chouinard et al., 2023). A previous study has shown that drainage or peatland-to-cropland conversion during the years 850-2010 could increase total historical carbon stocks in northern peatlands by 72 Pg C (Qiu et al., 2021). However, because the development presented in this study is primarily aimed at projecting future changes in deep carbon stored in peatlands. Omitting historical carbon losses from drainage is likely to have a relatively limited influence on future projections, as those losses mostly contributed to past emissions.**"

3. *This is a publication is submitted now, in 2025, but most of the references in the introduction are out of date and, even more importantly, the datasets used in the analysis and as model inputs are also out of date. These need to be updated to include the most recent research. This would be desirable in any case but I think this is really imperative given major point #1. Consider, for example, the dataset presented in Treat et al. (2019) contains over 3900 basal ages of peatland initiation plus information on C accumulation in peatlands over time and a comparison between spatial extent and C stock. Hugelius et al. (2020) contains new information on peatland C stocks. Mishra et al. (2021) has additional analysis of permafrost C stocks. Strauss et al. (2017) and Strauss et al. (2021) have updated information about Yedoma stocks. Treat et al. (2021) has information on spatial coverage and evolution of northern peatlands over time. Brosius et al. (2021) include lake initiation ages for Yedoma thermokarst lakes, which is in some confusing way alluded to in the description of peatlands. Kleinen et al. (2012) and 2016 also describe peatland initiation and C storage development during the*

*Holocene, which has also been done empirically by Nichols & Peteet (2019). The list goes on, these are only suggestions, but an improved literature review is needed.*

**[Response]** Thank you again for your insightful comment. We agree that the references in the Introduction section were missing some recent publications, and we have updated them accordingly. However, we respectfully disagree with the suggestion that the datasets used in our analysis are out of date. We have used what we believe to be the most current and relevant sources available for our study. Specifically, Yedoma and peatlands are two primary components of our development. For Yedoma, we used the Yedoma map from Strauss et al. (2021), which to our knowledge is the most recent Yedoma dataset and aligns with your suggestion. For peatland extent, we used the map from Hugelius et al. (2020), which is also one of the latest peatland area datasets. Therefore, we believe the datasets used in our analysis are not outdated. The maps of peat basal ages used to prescribe peat initiation are from Loisel et al (2017). While this reference does not contain all known peat basal core data, the data is provided by the oldest reliable peat inception ages, aggregated at a 1° × 1° resolution, which is well-suited for broad-scale modeling.

Regarding newer data of peat basal ages presented in Treat et al., (2019) and Treat et al., (2021), we have provided a detailed comparison in peat basal age and site numbers between the suggested dataset and the one used in our study in our following response to one of your minor comment #3. Please see more details below.

For an improved literature review, thank you for this helpful comment. Following your suggestion, in the revised Introduction section, we have updated the numbers and references for deep C stocks, including the extent of Yedoma deposits (Strauss et al., 2021) and their carbon stocks (Strauss et al., 2017), the extent and carbon stocks for peatlands (Hugelius et al., 2020), and estimates of 2-3 m deep permafrost carbon (Mishra et al., 2021). In the revised Methods section, we updated references for the development of Yedoma deposits (Strauss et al., 2017), and added new references for peatland development (Brosius et al., 2021; Treat et al., 2019, 2021). In the revised Discussion section, we included additional references for Holocene peatland carbon accumulation simulations (Kleinen et al., 2012, 2016).

**Minor comments:**

1. *Abstract*

    *Define deep yedoma deposits*

    *Define spatial extent of analysis*

    *What about CLM and earlier Orchidee where this is done for permafrost?*

    *Is C stocks all that matter?*

**[Response]** Thanks for this comment. We agree and have added the definition of Yedoma deposits **"(ice-rich, organic-rich permafrost, formed during the late Pleistocene)"** and the spatial extent of our analysis **(30°N-90°N)** in the revised abstract. Regarding your broader question: for permafrost regions, in addition to their substantial carbon stocks, changes in energy cycles (e.g., freeze–thaw processes of pore and ground ice) and hydrological cycles (e.g., thermokarst lake formation or drainage) with climate warming are also critical concerns. The development presented in our study primarily focuses on improving the representation of deep carbon stocks stored in Yedoma and peatlands, and will be applied in the next step to assess how these deep carbon pools influence the modeled response of permafrost to future climate warming. While not the only aspect of permafrost-climate feedback, this is a key research question in this field.

In CLM and earlier ORCHIDEE versions, simulations of permafrost carbon stocks are limited to soil carbon accumulation down to 3 m for conventional soils. For peatlands, the earlier ORCHIDEE model is capable of simulating carbon accumulation but assumes simultaneous peatland initiation across all northern regions (Qiu et al., 2019; Xi et al., 2024). In our study, we improved this by prescribing spatially explicit peatland inception times and locations using peat age maps, allowing a more realistic representation of northern peatland development.

2. *Intro: some of the numbers and references for deep C stocks in the north seem out of date (e.g. there are newer papers). Same with Yedoma. Same with tropical peatlands. Many of the cited references are over 10 years old, there are many new developments and accounting that has happened since! The only more recent one is Hugelius et al 2020!*

**[Response]** Thanks. Following our previous response, we have updated the numbers and references for deep C stocks in the revised Introduction. These updates include the extent of Yedoma deposits (Strauss et al., 2021) and their carbon stocks (Strauss et al., 2017), the extent and carbon stocks for peatlands (Hugelius et al., 2020), and estimates of 2-3 m deep permafrost carbon (Mishra et al., 2021).

3. *Methods peatlands*
   *Basal ages: Treat et al. 2019/2021 seems more appropriate with a focus on permafrost ecosystems.*

**[Response]** Thank you for this constructive suggestion. As explained above, a main motivation for originally choosing data from Loisel et al. 2017 was the fact that data is ordered by 1° × 1° degrees, suited for gridding. As suggested, we downloaded the peat age dataset compiled by Treat et al., (2021), and compared it with the dataset used in our study, from Loisel et al., (2017). As shown in Fig. R2, although the Treat et al. dataset contains 844 more sites than the Loisel et al. dataset, both datasets show highly consistent spatial patterns and frequency distributions of site-level peat age. When considering only unique sites, the number of valid records drops to 2,565 for Treat et al. and 2,267 for Loisel et al.

(Table R1), indicating a slightly higher proportion of replicas for the same site in Treat et al. dataset. The gridded mean peat age, calculated as the average of site ages within each grid cell, also shows a strong agreement between the two datasets (Fig. R3). Over 85% of grid cells show a peat age difference smaller than 1 thousand years, the interval of epochs / age cohorts of our peat simulation. In a few grid cells located in the southern region of North America, the peat age difference between the two datasets is more significant, but the peatland cover fraction in this area is smaller than 5% (Fig. S2(p)). Besides, the Loisel dataset shows slightly older gridded mean peat ages than the Treat et al. dataset in the southern region (Fig. R3(c)), but this difference is due to a greater number of sites in the Loisel dataset, rather than fewer (Fig. R3(f)), suggesting the number of sites in the Loisel dataset is not always smaller than in the Treat dataset at grid level.

In summary, although the dataset from Treat et al. (2021) includes 844 more total sites and 298 more unique sites, the overall spatial patterns and age distributions are quite similar. Therefore, we suspect that replacing the Loisel dataset with the Treat dataset would not change the reconstructed peat age maps too much. Given the high computation costs in running spin-up simulations spanning tens of thousands years, we kept our current results for the moment. However, for our future work, we will merge both datasets to further improve the reconstruction of Holocene peatland development.

[Figure]

Fig. R2. Comparison of peat age datasets between Loisel et al., (2017) and Treat et al., (2021) across the region 30°N-90°N. (a) and (b), Spatial distribution of peat ages from 2,850 sites (after removing 10

sites located south of 30°N and/or lacking valid records) in Loisel et al. (2017) and 3,694 sites (after removing 288 such sites in Treat et al. (2021), respectively. (c) and (d), Frequency distribution of site numbers across 15 age bins corresponding to datasets in (a) and (b), respectively.

Table R1. Total and unique site number for the region 30°N-90°N and its three subregions including North America, Europe, and Asia. Note that the sum of site numbers across the three subregions does not equal the total for the entire 30°N-90°N region, as some sites fall outside the defined continental boundaries.

| Site Number | | 30°N-90°N | North America | Europe | Asia |
|---|---|---|---|---|---|
| Total | Loisel et al., 2017 | 2,850 | 1,637 | 531 | 272 |
| | Treat et al., 2021 | 3,694 | 1,995 | 827 | 364 |
| Unique | Loisel et al., 2017 | 2,267 | 1,361 | 379 | 194 |
| | Treat et al., 2021 | 2,565 | 1,485 | 427 | 271 |

[Figure]

Fig. R3. Gridded comparison of peat age between Loisel et al., (2017) and Treat et al., (2021) across the region 30°N-90°N. (a) and (b), Spatial distribution of gridded peat age, calculated as the mean site age within each grid cell, from the Loisel and Treat datasets, respectively. (d) and (e), Spatial distribution of site number within each grid cell from the two datasets. (c) and (f), Difference in peat age ((a) minus (b)) and site number ((d) minus (e)) between the two datasets, respectively. Grid cells with zero difference are omitted from (c) and (f) to improve the visibility of differences.

4. *Addition of info on yedoma here is confusing. Why is the yedoma in the peatlands session? Why don't you use the Brosius et al dataset that looks at lake formation? Oh, it's just to justify peatlands, that yedoma turns off and peatlands turn on?*

**[Response]** Sorry for the confusion. We would like to clarify the two phases of the new spin-up simulation once again. First, we simulated Yedoma carbon accumulation formed in the late Pleistocene by turning on the sedimentation process for Yedoma deposits (Eq. (1)), using the sedimentation rate (1.0 mm yr⁻¹) and duration (19,000 years) inferred from the site-measured heights and ages of Yedoma deposits. Beginning at 13.5 ka, we transitioned to simulating carbon accumulation in Holocene peatlands and conventional soils. In this second simulation phase, we included 17 plant functional types (PFTs), with the 17th PFT representing Yedoma. We reintroduced Yedoma during this phase because our study is aimed to simulate Yedoma deposits that still exist today, that is, deposits that survived through the Holocene. To do this, we transferred the litter and soil carbon from the final year of the Yedoma simulation to initialize the carbon pools of the Yedoma PFT in the second simulation phase. However, we turned off sedimentation, letting the Yedoma PFT follow the same processes as conventional PFTs and accumulate carbon slowly. As shown in Fig. 5(d), although Yedoma carbon was reintroduced in the second simulation phase, we found that its carbon stock remains nearly stable throughout the Holocene. This suggests that our simulated energy budgets are consistent with the observation-based Yedoma distribution from Strauss et al. (2021), and our simulation protocol is both valid and effective.

In response to your comment, we have added more text in the Methods section to clarify why we reintroduced Yedoma in the peat simulation. "**We reintroduced Yedoma during this phase because our study is aimed to simulate Yedoma deposits that still exist today, that is, deposits that survived through the Holocene. To do this, we transferred the litter and soil carbon from the final year of the Yedoma simulation to initialize the carbon pools of the Yedoma PFT in this phase. However, we turned off sedimentation, letting the Yedoma PFT follow the same processes as conventional PFTs and accumulate carbon slowly.**" (L277-281)

Regarding lake formation, we did not use lake age / sediment datasets as our current model does not yet include lake-related processes, in particular for thermokarst lakes. However, we have started working on representation of ground ice and the formation of thermokarst lakes in our model version. Many thanks for suggesting this dataset from Brosius et al. It's very helpful for our upcoming model developments.

5. *Methods Yedoma/peatland is also a bit tricky, there is this Holocene cover on top of Yedoma deposits that can be 1-2m thick that I can't quite figure out how is dealt with in this modeling approach. This is much more C rich than Yedoma.*

**[Response]** As explained in our previous response, the Yedoma PFT still exists during the peatland simulation stage. Due to increased vegetation productivity under the warmer Holocene climate compared to the Last Glacial Maximum, we do find that the simulated carbon density for upper soil layers (0-2 m) of Yedoma PFT is slightly higher at the end of the second simulation stage (Fig. 7(g)) than at the end of the first simulation stage (Fig. 4(c)). In addition, our simulated increase in total Yedoma carbon stock through the Holocene (16 Pg C) matches well with the estimates (13 ± 1) from Strauss et al., (2017). In the revised manuscript, we have added this evaluation in Section 3.2. "**Due to increased vegetation productivity under the warmer Holocene climate compared to the LGM, the simulated Yedoma carbon stock increases by 16 Pg C through the Holocene, which matches well with the estimates of 13 ± 1 Pg C from Strauss et al., (2017).** " (L367-369)

6. *Results*

   *Yedoma, what about Jens synthesis?*

   *Super high C densities, is this really considering ice density and ice and ice wedges? Seems higher than estimates that I've seen from Hugelius et al. which could be an average.*

**[Response]** In the synthesis by Jens Strauss and colleagues, the total carbon stocks for Yedoma deposits are estimated at 115 (83-269) Pg C (Strauss et al., 2017), in which range our estimate of 141 Pg C falls. Regarding vertical carbon density, we would like to emphasize the importance of units. While values such as 271 kg C m$^{-2}$ and 366 kg C m$^{-2}$ may be super high, it is important to recognize that they represent cumulative carbon across the ~20 m depth of the Yedoma soil column. When examining our simulated volumetric carbon density of ~14 kg C m$^{-3}$ when accounting for ice wedges (or ~28 kg C m$^{-3}$ if we exclude ~50% of the soil column volume due to ground ice), our results align well with previous estimates, such as 25.98 ± 1.5 kg C m$^{-3}$ (excluding ice) reported by Anthony et al. (2014), and 15.4-26.8 kg C m$^{-3}$ (excluding ice) reported by Palmtag et al. (2022). These comparisons are already included in our original manuscript.

7. *Peat PFT is a bit high, check temporal trends, seems too little in the initial stages.*

**[Response]** Compared to previous estimates of northern peatland carbon stocks, e.g., 547 (±74) Pg C from Yu et al. (2010), 415 ± 150 Pg C from Hugelius et al. (2020), 410 (315-590) Pg C from Treat et al. (2019), and 463 Pg C from the earlier ORCHIDEE-MICT-PEAT version in Qiu et al. (2019), our simulated value (~630 Pg C) is indeed at the high end. This overestimation may be attributed to the inclusion of SOC-soil temperature coupling at the PFT level in our model version. As evaluated in our previous work (Xi et al., 2024), wetter, high-SOC peat soils result in lower soil temperatures compared to the grid-cell average. When this coupling is implemented at the PFT level (i.e., tiled coupling), it leads to a higher simulated peat SOC stock of 620 Pg C compared to 534 Pg C under grid-cell-level coupling (closer to previous estimates), i.e., an increase of 86 Pg C. This finding suggests that further evaluation and calibration of peat SOC in the improved model version are still needed. However,

because the main focus of this study is on the impact of spatially varying peatland initiation on long-term peat carbon accumulation, we consider such calibration work to be beyond the scope of this paper. In response to your comment, we have added this discussion to Section 4.1 to acknowledge this overestimation in peatland carbon stocks.

"**Regardless of whether peatlands are initiated simultaneously or spatially variably, the simulated total northern peatland carbon stock (~630 Pg C) is at the high end compared to previous estimates, e.g., 547 ± 74 Pg C from Yu et al. (2010), 415 ± 150 Pg C from Hugelius et al. (2020), 410 (315–590) Pg C from Treat et al. (2019), and 463 Pg C from the earlier ORCHIDEE-MICT-PEAT version in Qiu et al. (2019). This overestimation may be also attributed to the SOC-$T_{soil}$ feedback (or coupling). As evaluated in Xi et al. (2024), wetter and higher-SOC peat soils tend to maintain lower temperatures than the grid-cell average. When SOC-$T_{soil}$ coupling is implemented at the PFT level (i.e., tiled coupling) in the base model (ORCHIDEE-MICT-teb), the simulated peat SOC stock increases to 620 Pg C, compared to 534 Pg C with grid-cell-level coupling, i.e., an 86 Pg C difference. This suggests that further evaluation and calibration of peat SOC in the improved model version are still needed in the future.**" (L501-509).

Regarding the temporal trend, the low total peatland carbon stock in the initial stages is due to the small peatland area (Fig. 3). Compared to the carbon stocks reported by Treat et al. (2019) for the same period, our estimates are much lower, which may result from discrepancies in how carbon stocks were calculated between the two studies. Treat et al. (2019) estimated total carbon stocks by multiplying peatland areas (simulated using TOPMODEL) by peat depth estimates and a mean carbon density, while our study uses a process-based representation of peatland accumulation and decomposition, along with peat-age-reconstructed peatland extent. We did not include a detailed comparison in the revised manuscript, as we consider our approach (based on process representation and observational constraints) to offer a more mechanistic and internally consistent estimation than the scaling approach used in Treat et al. (2019).

8. *Difference in spatial domains is really confusing. Only from Table 3 does it become clear that the peat simulation extends to 30 N but the Yedoma simulation is clearly not relevant here.*

[Response] As clarified in our response to your major comment #2, the spatial domain for the Yedoma simulation is 60°N-90°N, while the peatland simulation spans a broader domain of 30°N-90°N.

**References mentioned by the reviewer**

Brosius, L.S., K.M. Walter Anthony, **C.C. Treat**, J. Lenz, M.C. Jones, M.S. Bret-Harte, G. Grosse (2021). Spatiotemporal patterns of northern lake formation since the Last Glacial Maximum. *Quaternary Science Reviews* 253, 106773. doi: 10.1016/j.quascirev.2020.106773

Hugelius, G., J. Loisel, S. Chadburn, R. B. Jackson, M. Jones, G. MacDonald, M. Marushchak, D. Olefeldt, M. Packalen, M. B. Siewert, **C. Treat**, M. Turetsky, C. Voigt and Z. Yu (2020). Large stocks of peatland carbon and nitrogen are vulnerable to permafrost thaw. *Proceedings of the National Academy of Sciences*: 201916387. doi: 10.1073/pnas.1916387117

Mishra, U., G. Hugelius, E. Shelef, Y. Yang, J. Strauss, A. Lupachev, J. Harden, J. Jastrow, C.-L. Ping, W. Riley, E.A.G. Schuur, R. Matamala, M. Siewert, L. Nave, C. Koven, M. Fuchs, J. Palmtag, P. Kuhry, **C. Treat**, S. Zubrzycki, F. Hoffman, B. Elberling, P. Camill, A. Veremeeva, A. Orr (2021). Spatial heterogeneity and environmental predictors of permafrost region soil organic carbon stocks. *Science Advances* 7 (9): eaaz5236. doi: 10.1126/sciadv.aaz5236.

Kleinen, T., Brovkin, V., and Schuldt, R. J.: A dynamic model of wetland extent and peat accumulation: results for the Holocene, Biogeosciences, 9, 235–248, https://doi.org/10.5194/bg-9-235-2012, 2012.

Kleinen, T., Brovkin, V., and Munhoven, G.: Modelled interglacial carbon cycle dynamics during the Holocene, the Eemian and Marine Isotope Stage (MIS) 11, Clim. Past, 12, 2145–2160, https://doi.org/10.5194/cp-12-2145-2016, 2016.

Treat, C. C., T. Kleinen, N. Broothaerts, A. S. Dalton, R. Dommain, T. A. Douglas, J. Z. Drexler, S. A. Finkelstein, G. Grosse, G. Hope, J. Hutchings, M. C. Jones, P. Kuhry, T. Lacourse, O. Lähteenoja, J. Loisel, B. Notebaert, R. J. Payne, D. M. Peteet, A. B. K. Sannel, J. M. Stelling, J. Strauss, G. T. Swindles, J. Talbot, C. Tarnocai, G. Verstraeten, C. J. Williams, Z. Xia, Z. Yu, M. Väliranta, M. Hättestrand, H. Alexanderson and V. Brovkin (2019). "Widespread global peatland establishment and persistence over the last 130,000 y." Proceedings of the National Academy of Sciences: 201813305.

Treat, C.C., Jones, M.C., Brosius, L., Grosse, G., Anthony, K.W. and Frolking, S., 2021. The role of wetland expansion and successional processes in methane emissions from northern wetlands during the Holocene. Quaternary Science Reviews, 257, p.106864.

*The new model setup departs from traditional, static assumptions used in conventional land surface model spin-ups. Instead, it integrates process-based, historically grounded reconstructions of deep carbon formation, resulting in a more realistic initialization of high-latitude carbon stocks.*

*Initialization and spin-up procedures are often underappreciated aspects of land surface modeling, yet they are critical for improving the accuracy of permafrost carbon feedback projections. This study represents an important and timely contribution to the field, highlighting the complexity of SOC modeling in high-latitude environments. It is well-written and comprehensive, and I have no further comments. I recommend this manuscript for immediate publication.*

**[Response]** Thank you very much for taking the time to review our manuscript and for your encouraging comments. We greatly appreciate your recognition of the significance and timeliness of our work.

Following the suggestions from the other two reviewers, we have further refined the manuscript by clarifying our model development and evaluation strategies. We have also expanded the model description to include more detail on changes in plant and soil carbon pools associated with the

conversion from conventional soils to peatlands. In addition, we have added further discussion on the limitations of our approach.

We believe these revisions have significantly improved the clarity and comprehensiveness of the manuscript. Thank you again for your thoughtful review!

---

## Author Response (AR2)

**Response to the editor**

*Dear Yi,*

*your revised manuscript has now been seen by another reviewer who has some additional comments. Please respond to the comments and also make sure that you follow the GMD policy that model code and data has to be archived in a persistent repository and that the version number is included in the title.*

*Best*

*Marko*

**[Response]** Thank you very much for your time and effort in handling our revised manuscript. We have carefully addressed the reviewer's additional comments and revised the manuscript accordingly. In compliance with the GMD policy, we have archived the model code in a public Zenodo repository: https://zenodo.org/records/15306029, and have archived all datasets used in the study, along with the processing scripts and code for generating the results and figures, in a separate Zenodo repository: https://zenodo.org/records/15371113. Both repositories are cited in the manuscript to ensure full reproducibility. Additionally, we have included the model version number in the title, as required.

**Response to the reviewer 4**

**General comments**

*Peatlands are one of the largest C reservoirs on land that actively interacts with the atmosphere. Peats as organic soils (histosols) are often much deeper than mineral soils types and have very different formation process due to waterlogging conditions. So more realistic treatment of peatlands is critical. Also, most organic carbon in Yedoma deposits have not been incorporated through soil forming processes such as roots or pedoturbation. The study incorporates and integrated these two "special" land surface units (peatlands and Yedoma deposits) using a new spin-up scheme into a popular land surface model (ORCHIDEE). This represents an important progress towards modeling deep land carbon for many applications in the future.*

*I have read the revised manuscript and authors' responses to previous round of reviewers' comments. In general, I think the manuscript is much improved.*

**[Response]** Thank you very much for taking the time to review our revised manuscript and for your encouraging feedback on both the manuscript and our previous responses. We sincerely appreciate your detailed, thoughtful, and constructive comments, which have helped us improve the clarity of the paper greatly. We have carefully revised the manuscript following your suggestions. Please find our point-by-point responses to your major and detailed comments below.

**Major comments**

*I do have a few additional questions that the authors may wish to consider.*

1. *When describing Yedoma deposits, the authors refer them either as Yedoma deposits or soils. During the late Pleistocene when the Yedoma was actively formed, it should be referred as Yedoma deposits, as there was almost no soil formation process. Most or all organic matter was buried through sedimentation processes, such as aeolian (wind-blown), alluvial (by water), colluvial (by gravity) or nival processes (through snow/meltwater). However, during the Holocene, Yedoma deposits are presumably no longer actively accumulating, so soils could be formed on these deposits. For the purpose of this study, in most occasions it should refer as Yedoma deposits. I feel that the authors should clarify this distinction.*

**[Response]** Thank you for this thoughtful suggestion. Our model was indeed designed to reflect the distinction between "Yedoma deposits" and "Yedoma soils": during the late Pleistocene, carbon accumulates in Yedoma deposits through the sedimentation processes. In the Holocene, these processes are turned off, allowing conventional soil formation and carbon accumulation to occur on top of the existing deposits. As our study focuses more on the carbon accumulation processes during the late Pleistocene, we now consistently use the term "Yedoma deposits" throughout the revised manuscript, as suggested by the reviewer.

2. *During the Yedoma formation period before the Holocene, there should be multiple PFTs that had grown on these deposits from the boreal to arctic regions when the weather/climate permits, as perhaps implemented in Zhu et al. (2016), despite that the organic matter did not necessarily enter the "soil" profile through pedoturbation or other soil forming processes, but instead through burial and sedimentation processes. Perhaps during the Holocene, the process should be similar to other conventional PFTs (see comment above). On that point, I'm not sure what does the term fi(z,t) (that is, "the OC input to pool i") in equation 2 refers to: burial/sedimentation as for Yedoma deposits or root or litter input as for soils on Yedoma deposits parent materials.*

**[Response]** In our simulation of Yedoma formation during the late Pleistocene, we use the dynamic global vegetation model in ORCHIDEE-MICT, as in Zhu et al. (2016), to allow multiple PFTs to grow wherever climate conditions permit. These PFTs provide organic matter, which is buried directly via sedimentation processes (as represented in Equation 2), rather than being incorporated through soil-forming mechanisms like pedoturbation. At the beginning of the Holocene, we transition to a single Yedoma PFT that inherits the total carbon previously accumulated by all PFTs in the Yedoma deposits. From that point on, sedimentation processes are turned off, and conventional soil formation continues on top of the existing deposits. This setup enables us to distinguish the future response of Yedoma-affected regions from that of conventional vegetation types like trees and grasses, as well as peatland PFT.

Regarding Equation 2: the term $fi(z, t)$ refers to the input of organic carbon (OC) into pool $i$, typically from root and litter inputs. In conventional soils, these inputs are incorporated into the soil profile and transported downward through soil-forming processes. By contrast, during Yedoma formation, the same organic inputs are buried through sedimentation rather than pedogenic processes. This burial is captured by the sedimentation term "$u(t)\frac{\partial C_i(z,t)}{\partial z}$" in Equation 2.

3. *Why the very different peatland initiation scheme/timings in the old and new spin-ups generate very similar total C stocks in peatlands? Are these simply coincidence or constrained by mass conversation or the net result of compensation of different peat C processes (such as NEP, decomposition, and various decomposition rates of different C pools)?*

**[Response]** Thank you for this great question. We also initially expected notable differences in total peatland carbon stocks, given that the new spin-up, which incorporates observation-constrained peatland initiation timing, substantially shortens the duration of peat carbon accumulation, particularly for peatlands younger than 13,500 years (Fig. S1). However, the total carbon stocks in peatlands remain similar between the old and new spin-ups. This is primarily due to the mass conservation constraint in ORCHIDEE-MICT, which we have explained in Section 4.1.

First, we do find that the shorter accumulation period in the new spin-up leads to a reduction of 20 Pg C (7%) in deep (>3 m) peat carbon and a 35 Pg C (43%) decrease in the passive SOC pool (the slowest-turnover carbon pool) (Table 3), particularly in younger peatlands (Fig. 6). Peat depths also become 1–5 m shallower, especially in the Boreal region (Fig. S5f), where total peat SOC decreases by 53 Pg C (18%) in the new spin-up compared to the old.

However, this loss is largely compensated by an increase in Arctic peatland carbon stocks, 49 Pg C (15%) higher in the new spin-up. This compensation arises from the model's mass conservation requirement: as peat PFTs expand during the Holocene, SOC from shrinking conventional PFTs (e.g., trees and grasses) is redistributed to the expanding peat PFTs. In Arctic regions, where conventional PFTs already store substantial SOC (up to 3 m depth; see Fig. 7 and Fig. S4), this inherited SOC results in even higher stocks and deeper profiles in the new spin-up (Fig. S5). By contrast, mid-latitude peatlands, where conventional PFTs store less SOC, receive less inherited carbon, leading to fewer disruptions to the impacts of constrained peatland initiation timing on peat carbon accumulation.

In summary, while the model's mass conservation constraint drives a regional compensation in total peat carbon stocks, the revised peatland initiation timing still plays a critical role in shaping the vertical distribution and turnover characteristics of peat carbon pools, especially in the Boreal region. The reductions in deep SOC, passive SOC, and peat depth under the new spin-up may alter the future vulnerability and response of peat carbon to climate change.

**Some detailed comments:**
*I have some editorial edits and suggestions to help improve the clarity of the manuscript. Specific comments:*

*Line 19: change to "… peatlands (formed mostly during the Holocene)…"*
**[Response]** Done.

*L23: not soil layer in Yedoma deposits. See comment above.*
**[Response]** Thanks. We have revised the sentence as "... that deposited deep carbon in **the layers of peatlands and Yedoma deposits.**".

*L62-63: perhaps better called it "deposits", rather than "sediments"*
**[Response]** Revised as suggested.

*L68: not Yedoma soils, but Yedoma deposits*

**[Response]** Revised.

*L108: Yedoma deposits*
**[Response]** Revised.

*L145: change "sedimentation" to ""sedimentation"" (with quotation marks), as it is an accumulation process by plants living on peatlands, not really sedimentation process*
**[Response]** Revised as suggested.

*L160: change to "downward movement rate"?*
**[Response]** Revised.

*L165: "Strauss et al. (2013)"*
**[Response]** Revised.

*L216 Table 1: Change "period" to "Duration"?*
**[Response]** Revised as suggested.

*L238: at the end add "smaller ... extent of Yedoma deposits south of 60 N"*
**[Response]** Thank you. We have revised the sentence slightly beyond the suggestion as: "**…given that no Yedoma deposits are found south of 60°N (Fig. 2; Strauss et al., 2021).**" According to the Yedoma deposit map in Strauss et al. (2021), "no" is more accurate than "smaller", as there is no documented extent south of 60°N.

*L242: add "," after land surface model"*
**[Response]** Added.

*L244 Table 2: change "Epoch" to "Period" and "Period" to "Duration" on both table heading lines. "Epoch" is official chronological term in geology stratigraphy, such as "Holocene Epoch".*
**[Response]** Revised as suggested.

*L260: change "benthic mosses" to "mosses"*
**[Response]** Done.

*L268 and elsewhere in this paragraph and in the manuscript: change "epochs" to "periods". See comment above.*
**[Response]** Revised throughout the manuscript.

*L271: change "...time in kyrs from 16 to 9" to "...time from 16 to 9 kyrs"*

**[Response]** Done.

*L290: Figure 3a: vertical axis label units to "Number per kyr" and "Mkm2 per kyr"*

**[Response]** Revised as suggested.

*L298 and line 304: change to "Xi et al. (2014)" (no comma)*

**[Response]** Done.

*L307: change "forest" to "tree"*

**[Response]** Revised.

*L316: "Results and evaluation" is a little weird. Change to "Results and interpretations"?*

**[Response]** Thanks for the suggestion. We have revised the heading as "**Results and interpretation**".

*L340: change to "kg per m2". Also, "...depth of total organic carbon" is not an accurate expression. Change to "...depth of Yedoma deposits" or "...depth of OC-rich Yedoma deposits"? Deposits have depth, but OC doesn't.*

**[Response]** Revised to "kg per m2" and "...depth of Yedoma deposits".

*L343-344: see comment on line 238. Add "south of 60 N"*

**[Response]** Following our response to your comment on line 238, we revised the sentence as: "**…given that no Yedoma deposits are found south of 60°N (Fig. 2; Strauss et al., 2021).**"

*L345: change "peat" to "peatlands"*

**[Response]** Done.

*L351-352: change "for" to "from conventional PFTs…", and also elsewhere?*

**[Response]** Revised throughout the manuscript.

*L355: in the figure panel (b) and (d) X-axis label: change "Simulation year" to "Simulation time"?*

**[Response]** Done.

*L364: change to "accumulate a total soil carbon stock of 1239 Pg C…"*

**[Response]** Done.

*L365: to "present-day Yedoma extent"*

**[Response]** Done.

*L379: "in the new spin-up than the old spin-up"*

**[Response]** Done.

*Table 3:*

*Left column: change to "Total OC stock", to "By depth interval (m)"*

**[Response]** Revised as suggested.

*L417 Figure 7 caption: these are not "soil types". Change to "comparison of simulation results from the old and new spin-ups"?*

**[Response]** Thank you for pointing this out. We have revised the caption as "Carbon accumulation for three **PFT classes** after the peat simulation from the old and new spin-ups." While we appreciate the suggested caption, we felt that "comparison of simulation results from the old and new spin-ups" was too general and did not clearly convey the figure's focus on PFT-specific carbon accumulation.

*L460: The 4.1 heading sounds too similar to "Results and evaluation/interpretation" section. Should focus on some more integrated and more important topics to discuss, rather than still on differences and interpretations of new and old spin-ups…*

**[Response]** Thanks for the suggestion. We have revised the heading to a more focused title: "**Effects of SOC insulation on soil carbon accumulation**".

*L504: The discussion on the SOC-Tsoil feedback is really good! It would be really useful to partition of relative contributions of different processes and mechanisms, including peat-T feedback, partition of OC pools (passive vs slow vs fast)/turnover times, perhaps in future analysis.*

**[Response]** Thank you for your positive feedback. We also consider this an important direction for our future analysis.

*L548-449: not only for peatlands. If the simulations use realistic paleoclimate input data sets, you would expect changes in PFTs at different time during the last 20 kyrs, even if there were no feedbacks between vegetation and climate through online simulations with ESM. This limitation should perhaps be mentioned here.*

**[Response]** Thank you for this helpful suggestion. We have mentioned this point in the revised discussion "Coupling with paleoclimate models (e.g., Kleinen et al., 2012, 2016; Treat et al., 2019) or using available datasets such as TraCE-21ka (He et al., 2013), **along with incorporating the dynamic**

**evolution of conventional PFTs (e.g., trees, grasses, and croplands) over the last 20,000 years, would enable a more realistic simulation of carbon accumulation in peatlands and conventional soils.**" (L548-551)

*L554-555: incomplete sentences. Change to "…in peatlands, omitting…"*
**[Response]** Revised as suggested.

*L559-560: …presents a new spin-up protocol…by representing deep carbon …. Peatlands more realistically in the ORCHIDEE,,,"*
**[Response]** Revised as suggested.

*L563: change to "after observation-constrained peat initiation time…"*
**[Response]** Revised.

*L566: "226 Pg larger that the old…"*
**[Response]** Revised.

*L563-564: "a model version for simulation of deep land carbon….transition and for projection of land carbon response to future warming"*
**[Response]** Thank you. We have revised this sentence as suggested.

**References**

Strauss, J., Laboor, S., Schirrmeister, L., Fedorov, A. N., Fortier, D., Froese, D., Fuchs, M., Günther, F., Grigoriev, M., Harden, J., Hugelius, G., Jongejans, L. L., Kanevskiy, M., Kholodov, A., Kunitsky, V., Kraev, G., Lozhkin, A., Rivkina, E., Shur, Y., Siegert, C., Spektor, V., Streletskaya, I., Ulrich, M., Vartanyan, S., Veremeeva, A., Anthony, K. W., Wetterich, S., Zimov, N., and Grosse, G.: Circum-Arctic Map of the Yedoma Permafrost Domain, Front. Earth Sci., 9, https://doi.org/10.3389/feart.2021.758360, 2021.

Zhu, D., Ciais, P., Krinner, G., Maignan, F., Jornet Puig, A., and Hugelius, G.: Controls of soil organic matter on soil thermal dynamics in the northern high latitudes, Nat. Commun., 10, 3172, https://doi.org/10.1038/s41467-019-11103-1, 2019.